# *PTPRG* is an ischemia risk locus essential for HCO₃⁻-dependent regulation of endothelial function and tissue perfusion

Kristoffer B Hansen[1], Christian Staehr[1], Palle D Rohde[2], Casper Homilius[1], Sukhan Kim[1], Mette Nyegaard[1], Vladimir V Matchkov[1], Ebbe Boedtkjer[1]*

[1]Department of Biomedicine, Aarhus University, Aarhus, Denmark; [2]Department of Chemistry and Bioscience, Aalborg University, Aalborg, Denmark

**Abstract** Acid-base conditions modify artery tone and tissue perfusion but the involved vascular-sensing mechanisms and disease consequences remain unclear. We experimentally investigated transgenic mice and performed genetic studies in a UK-based human cohort. We show that endothelial cells express the putative HCO₃⁻-sensor receptor-type tyrosine-protein phosphatase RPTPγ, which enhances endothelial intracellular $Ca^{2+}$-responses in resistance arteries and facilitates endothelium-dependent vasorelaxation only when $CO_2$/HCO₃⁻ is present. Consistent with waning RPTPγ-dependent vasorelaxation at low [HCO₃⁻], RPTPγ limits increases in cerebral perfusion during neuronal activity and augments decreases in cerebral perfusion during hyperventilation. RPTPγ does not influence resting blood pressure but amplifies hyperventilation-induced blood pressure elevations. Loss-of-function variants in *PTPRG*, encoding RPTPγ, are associated with increased risk of cerebral infarction, heart attack, and reduced cardiac ejection fraction. We conclude that *PTPRG* is an ischemia susceptibility locus; and RPTPγ-dependent sensing of HCO₃⁻ adjusts endothelium-mediated vasorelaxation, microvascular perfusion, and blood pressure during acid-base disturbances and altered tissue metabolism.

*For correspondence:
eb@biomed.au.dk

**Competing interests:** The authors declare that no competing interests exist.

## Introduction

Inadequate tissue perfusion relative to metabolic demand is a fundamental pathophysiological cause of acute (e.g. myocardial infarction and stroke) and chronic (e.g. heart failure and neurodegenerative disorders) cardiovascular disease. Newer treatment options improve the prognosis for some of these debilitating conditions but we need alternative therapeutic strategies in order to minimize ischemia-related morbidity and mortality.

Sensing of the chemical environment in the wall of arteries coordinates local blood flow to meet the oxidative metabolic demand and counteract ischemia. Despite their obvious clinical importance, the cellular and molecular mechanisms responsible for sensing metabolic disturbances and respond-ing to inadequate perfusion are not well understood (*Boedtkjer, 2018*). Local acidification is an important signal of insufficient nutrient delivery and waste product elimination. Although functional effects of $H^+$ in the vascular wall have long been appreciated (*Boedtkjer, 2018*; *Gaskell, 1880*; *Boedtkjer et al., 2016a*; *Boedtkjer and Aalkjaer, 2012*; *Boedtkjer and Aalkjaer, 2013a*; *Boedtkjer and Aalkjaer, 2013b*), the consequences of associated changes in $CO_2$/HCO₃⁻ buffer composition have received far less attention. $CO_2$ and HCO₃⁻ constitute an important buffer pair that minimizes acute changes in pH (*Roos and Boron, 1981*), improves spatial $H^+$ mobility (*Spitzer et al., 2002*; *Boedtkjer et al., 2016b*), and provides substrate for acid-base transporters in cell membranes (*Aalkjaer and Hughes, 1991*; *Boedtkjer et al., 2006*). Evidence supports that cells also possess sensors that respond to changes in [HCO₃⁻] in the intracellular and extracellular space (*Zhou et al., 2016*; *Boedtkjer et al., 2016c*; *Chen et al., 2000*). Interventions—such as NaHCO₃

**eLife digest** Restricted blood flow in the heart or brain can deprive these vital organs of oxygen, thereby causing a heart attack or stroke. However, the body has compensatory mechanisms to mitigate damage: if the blood flow is reduced in one blood vessel, acidic waste accumulates locally. This causes nearby blood vessels to widen and increase the oxygen supply. Although scientists first observed this process 140 years ago, they have not yet devised a way to use it for treatment of heart attack or stroke.

Now, Hansen et al. discovered that a protein called RPTPγ, which is found on the lining of blood vessels, could be a good target for drugs intended to reduce the consequences of heart attacks and strokes. The protein RPTPγ has a similar structure to other proteins that bind bicarbonate, an important ion that buffers acids in the body. RPTPγ can also trigger signals to nearby cells, which suggests that the protein can monitor bicarbonate levels in the blood and tissue and alert blood vessels of the need to widen.

Hansen et al. found that the blood vessels of mice that lacked RPTPγ were unable to widen when needed. Moreover, mice without RPTPγ experienced abnormal changes in blood pressure and blood flow to the brain when oxygen consumption was elevated or pH was disrupted. Hansen et al. further analyzed genetic and health data from nearly 50,000 individuals in the UK Biobank. These analyses revealed that people with genetic changes that render RPTPγ ineffective are at higher risk of having a heart attack or stroke. People with these genetic variants also have reduced heart pumping ability.

The experiments suggest that a lack of functional RPTPγ affects an individual's ability to adjust local blood flow in response to acid-base disturbances and oxygen deficits, increasing the risk of a heart attack or stroke. This information may help scientists develop new ways to prevent or treat heart attacks and strokes, or to treat other conditions like cancer, where pH is disturbed.

supplementation—that modify systemic acid-base status are available (*Voss et al., 2020*) but expected to carry considerable adverse effects and be too general for cardiovascular therapy. The identification of $HCO_3^-$-related proteins (e.g. carbonic anhydrases, $Na^+,HCO_3^-$-cotransporters, $Cl^-/HCO_3^-$-exchangers, and $HCO_3^-$-sensors) opens largely unexplored avenues for pharmacological treatment. Because the vasculature of vital organs—particularly the brain and heart—responds to metabolic deregulation and is sensitive to acid-base disturbances, it is likely that therapy targeting $HCO_3^-$-related proteins will be able to modify myocardial and cerebral perfusion preferentially in areas of unmet metabolic demand.

Through mechanisms that require the transmembrane Receptor Protein Tyrosine Phosphatase (RPTP)γ, isolated decreases in extracellular $[HCO_3^-]$ enhance $HCO_3^-$ reabsorption in the renal proximal tubule (*Zhou et al., 2016*) and contractions of basilar arteries (*Boedtkjer et al., 2016c*) when $pCO_2$ and pH are kept constant using out-of-equilibrium technology (*Zhao et al., 1995*). The extracellular domain of RPTPγ resembles the active site of the carbonic anhydrases (*Barnea et al., 1993*) and therefore likely binds $HCO_3^-$. However, the carbonic anhydrase-like domain of RPTPγ lacks the histidine residues considered essential for catalyzing equilibration of the reaction: $CO_2 + H_2O \rightleftharpoons HCO_3^- + H^+$ (*Zhou et al., 2016*). Instead, $HCO_3^-$ may induce a dimerization-dependent auto-inhibitory response in RPTPγ and thereby regulate signaling via the intracellular tyrosine phosphatase domains (*Barr et al., 2009*). Apart from these structural observations, we do not currently understand the cellular and molecular mechanisms for impact of $HCO_3^-$ and RPTPγ on resistance artery function, blood pressure, and control of tissue perfusion. Although $CO_2$ has no direct net effect on cerebrovascular tone (*Boedtkjer et al., 2016c*), respiratory changes leading to hyper- or hypocapnia will lead to secondary rises or falls in $[HCO_3^-]$ and pH that in turn influence artery contractions and tissue perfusion.

In the present study, we demonstrate—based on experimental investigations in transgenic mice—that RPTPγ (a) facilitates endothelium-dependent relaxation of resistance arteries through mechanisms regulated by extracellular $HCO_3^-$ and (b) adjusts microvascular perfusion and blood pressure during increased tissue metabolism and acid-base disturbances. Through translational genetic studies in the UK Biobank cohort (*Bycroft et al., 2018*), we further substantiate the importance of

the identified mechanisms for human cardiovascular health as we demonstrate that predicted loss-of-function variants in *PTPRG*, encoding RPTPγ, are associated with human ischemic vascular disease in the heart and brain.

## Results

### *Ptprg* is widely expressed in the vascular endothelium

We first evaluated promoter activity for *Ptprg* by β-galactosidase staining of mice with a promoter-less *LacZ* insert under transcriptional control of the *Ptprg* promoter. We found signs of prominent *Ptprg* transcriptional activity in the endothelium of mouse basilar, middle cerebral, and coronary arteries (*Figure 1A*). We also saw signs of *Ptprg* transcriptional activity in pulmonary and skeletal muscle arteries, whereas transcription of *Ptprg* appeared lower in the aorta and mesenteric arteries (*Figure 1A*).

Since promoter activity is no direct measure of expression—and cytosolic dilution of the chromogenic reaction product lowers the sensitivity of the β-galactosidase reporter assay—we followed up with quantitative RT-PCR analyses that confirmed substantial steady-state levels of *Ptprg* mRNA in basilar, middle cerebral, coronary, and pulmonary arteries and identified clearly detectable levels in the aorta, mesenteric, and skeletal muscle arteries (*Figure 1B*).

We next explored the endothelial expression pattern of *Ptprg* based on single-cell transcriptomic data from healthy mice (*Kalucka et al., 2020*). These data verify *Ptprg* expression in the vascular endothelium of the brain, heart, lung, skeletal muscle, and small intestine (*Figure 1C,D*). The data further show that the endothelial *Ptprg* expression extends from the arterial vasculature, across capillaries, and into the veins (*Figure 1D,E*).

Taken together, we identify widespread endothelial *Ptprg* expression with prominent levels in the cerebral and coronary vasculature where metabolites, including acid-base equivalents, strongly influence arterial tone (*Gaskell, 1880*; *Boedtkjer et al., 2016c*; *Wahl et al., 1970*).

### RPTPγ enhances endothelium-dependent vasorelaxation

Isolated small arteries pre-contracted with the thromboxane A$_2$ analog U46619 relaxed concentration-dependently in response to the classical endothelium-dependent agonist acetylcholine (*Figure 2*). In order to evaluate a distinct additional endothelial signaling pathway (*D'Andrea et al., 1998*) capable of inducing potent endothelium-dependent vasorelaxation of cerebral and mesenteric small arteries (*Boedtkjer et al., 2013a*; *McNeish et al., 2005*; *Bucci et al., 2005*), we also tested the effect of the proteinase-activated receptor (PAR)2 agonist SLIGRL-amide (*Figure 3*).

Under physiological conditions with $CO_2/HCO_3^-$ present in the buffer solutions, acetylcholine- and SLIGRL-amide-induced vasorelaxation of basilar (*Figures 2A–C* and *3A–C*), mesenteric (*Figures 2I* and *3I*), and coronary (*Figures 2M* and *3L*) arteries from RPTPγ knockout (KO) mice was reduced compared to arteries from wild type (WT) mice. We observed this difference between arteries from RPTPγ KO and WT mice whether experiments in presence of $CO_2/HCO_3^-$ were conducted with (*Figures 2B* and *3B*) or without (*Figures 2C* and *3C*) the non-physiological buffer HEPES. Addition of an artificial buffer is necessary for pH control in absence of $CO_2/HCO_3^-$; and we included HEPES also in the $CO_2/HCO_3^-$-containing solutions in order to separate the vascular effects of $CO_2/HCO_3^-$ omission from potential effects of adding HEPES (*Altura et al., 1980*). Acetylcholine-induced vasorelaxation of basilar arteries from RPTPγ KO mice was suppressed in presence of $CO_2/HCO_3^-$ mostly because of attenuated NO-dependent (i.e. *N*-nitro-L-arginine methyl ester (L-NAME)-sensitive) signaling (*Figure 2D–G*). In contrast, the diminished vasorelaxation of basilar arteries in response to SLIGRL-amide (*Figure 3D–F*) and of mesenteric and coronary arteries in response to acetylcholine (*Figure 2J–L* and *Figure 2N–P*) and SLIGRL-amide (*Figure 3J,K,M,N*) was mostly explained by smaller endothelium-dependent hyperpolarization (EDH)-type responses that were L-NAME-insensitive but inhibited by apamin and TRAM-34. Consistent with these findings, L-NAME-insensitive smooth muscle hyperpolarization was significantly attenuated in basilar arteries from RPTPγ KO mice when elicited by SLIGRL-amide (*Figure 3G,H*) but not when elicited by acetylcholine (*Figure 2H*).

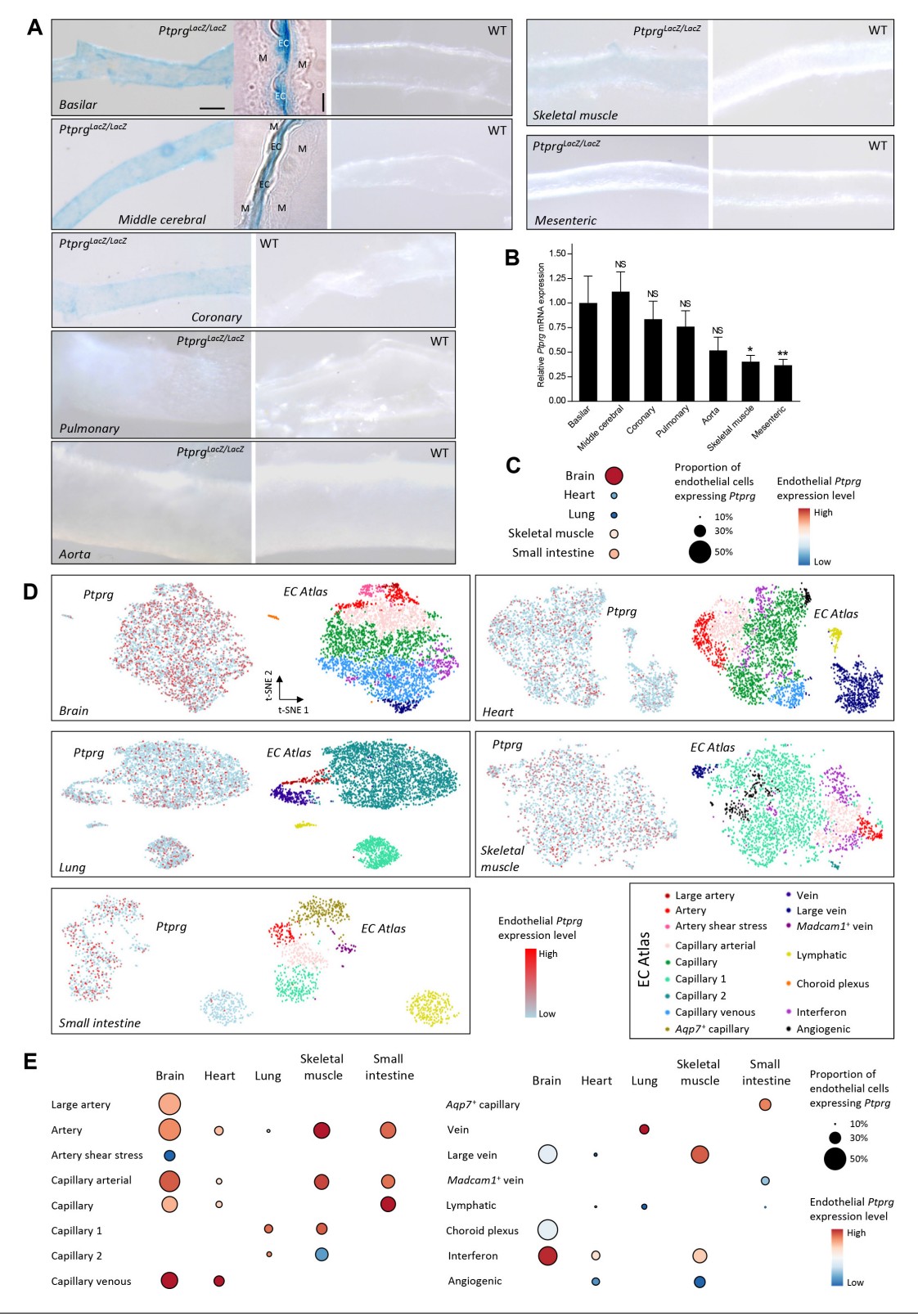

**Figure 1.** Profiles of *Ptprg* transcriptional activity and mRNA levels demonstrate widespread expression across arterial, capillary, and venous endothelial cells and in diverse vascular beds including cerebral and coronary arteries. (**A**) Histochemical staining for β-galactosidase activity reports the *Ptprg* transcriptional activity in *Ptprg^LacZ/LacZ* mice. The scale bars indicate 5 and 50 μm for the histological and whole-mount images, respectively, and all images within each category are displayed at the same magnification except for the image of the aorta where the scale bar corresponds to 150 μm. *Figure 1 continued on next page*

Figure 1 continued

The 4-µm-thick longitudinal histological sections illustrate basilar and middle cerebral arteries where the lumen is collapsed, and endothelial cells (EC) from opposing walls meet at the center and are surrounded by their adjacent tunica media (M). The images are representative of tissue from 4 *Ptprg$^{LacZ/LacZ}$* mice and 2 WT mice. (B) *Ptprg* mRNA levels in arteries from WT mice displayed relative to the ribosomal S18 and β-actin reference genes (n = 11) and normalized to the average level in basilar arteries. (C–E) Single-cell RNA sequencing data analyzed for endothelial *Ptprg* expression in healthy murine brain (3482 cells), heart (3219 cells), skeletal muscle (m. soleus, 2457 cells), lung (3972 cells), and small intestine (974 cells). Data are averaged across each vascular bed (C), displayed as t-SNE plots of *Ptprg* transcript patterns (left images) and clustered vessel types (right images) (D), and divided and quantified according to vessel type in each tissue (E). The single-cell RNA sequencing data are from the online EC Atlas database (*Kalucka et al., 2020*). Data in panel B were compared after log-transformation using repeated-measures one-way ANOVA followed by Dunnett's post-test. *p<0.05, **p<0.01, NS: not significantly different vs basilar arteries.

## The vasorelaxant influence of RPTPγ depends on $CO_2/HCO_3^-$

In order to evaluate whether RPTPγ-dependent signaling is influenced by $HCO_3^-$ in the vascular wall, we next repeated studies of endothelium-dependent vasorelaxation using nominally $CO_2/HCO_3^-$-free, HEPES-buffered solutions (*Figure 4*). Under these conditions, we saw no RPTPγ-dependent differences in overall vasorelaxant function of basilar or mesenteric arteries in response to acetylcholine (*Figure 4A,B,G*) or SLIGRL-amide (*Figure 4K,L,P*). The relative contribution of NO-dependent signaling and EDH-type responses to vasorelaxation was also not significantly affected by RPTPγ KO in basilar (*Figure 4C–F*) or mesenteric (*Figure 4H–J*) arteries in response to acetylcholine. Whereas the dependency on NO-signaling vs EDH-type vasorelaxation in response to SLIGRL-amide was likewise not affected in mesenteric arteries (*Figure 4Q,R*), we saw indications of an apparent relative increase in NO-mediated vasorelaxation when basilar arteries from RPTPγ KO mice were stimulated with SLIGRL-amide in absence of $CO_2/HCO_3^-$ (*Figure 4M–O*). This unexpected change in the relative importance of the underlying vasorelaxant mechanisms in basilar arteries could be a compensation secondary to the predominant inhibition of NO-dependent vasorelaxation in these arteries in presence of $CO_2/HCO_3^-$ (*Figure 2F*) and hence presumably in vivo.

## NO-synthase expression and responses to NO donors are largely independent of RPTPγ

We next explored the mechanism whereby RPTPγ influences resistance artery function (*Figure 5*).

We first evaluated whether the difference in vasorelaxation between arteries from RPTPγ KO and WT mice depends on a change in expression of the endothelial NO-synthase (eNOS). Based on immunoblotting of basilar (*Figure 5A*) and mesenteric (*Figure 5D*) arteries, we found equal eNOS expression between arteries from RPTPγ KO and WT mice.

We also evaluated whether the smooth muscle vasorelaxant responses to exogenous NO differ between arteries from WT and RPTPγ KO mice. In presence of $CO_2/HCO_3^-$, spermine NONOate and *S*-nitroso-*N*-acetylpenicillamine (SNAP) produced equivalent vasorelaxation in basilar and mesenteric arteries from RPTPγ KO and WT mice (*Figure 5B,C,E,F*).

Together, our findings support that RPTPγ exerts acute influences on endothelial signaling pathways regulating the activity of endothelium-dependent vasorelaxation in response to $[HCO_3^-]$. This interpretation is in agreement with the near-complete normalization of vasorelaxation observed in arteries from RPTPγ KO mice upon omission of $CO_2/HCO_3^-$ (*Figure 4*).

## RPTPγ enhances endothelial $Ca^{2+}$ responses with no effect on intracellular pH

We then determined whether endothelial intracellular $Ca^{2+}$ signals differ between arteries from RPTPγ KO and WT mice (*Figure 5G–N*). Loading of isolated arteries with $Ca^{2+}$-sensitive fluorophores through luminal perfusion ensured endothelium-specific fluorescence signals (*Figure 5G,K*).

In the presence of $CO_2/HCO_3^-$, application of acetylcholine elevated intracellular $[Ca^{2+}]$ substantially more in endothelial cells of basilar and mesenteric arteries from WT mice than in arteries from RPTPγ KO mice (*Figure 5H,J,L,N*). In contrast, we saw no significant difference between the intracellular $Ca^{2+}$ responses of endothelial cells in arteries from WT and RPTPγ KO mice when they were investigated in the nominal absence of $CO_2/HCO_3^-$ (*Figure 5I,J,M,N*).

Reduced endothelial intracellular $Ca^{2+}$ responses in arteries from RPTPγ KO mice in presence of $CO_2/HCO_3^-$ likely explain the observed mixed attenuation of NO-mediated and EDH-type

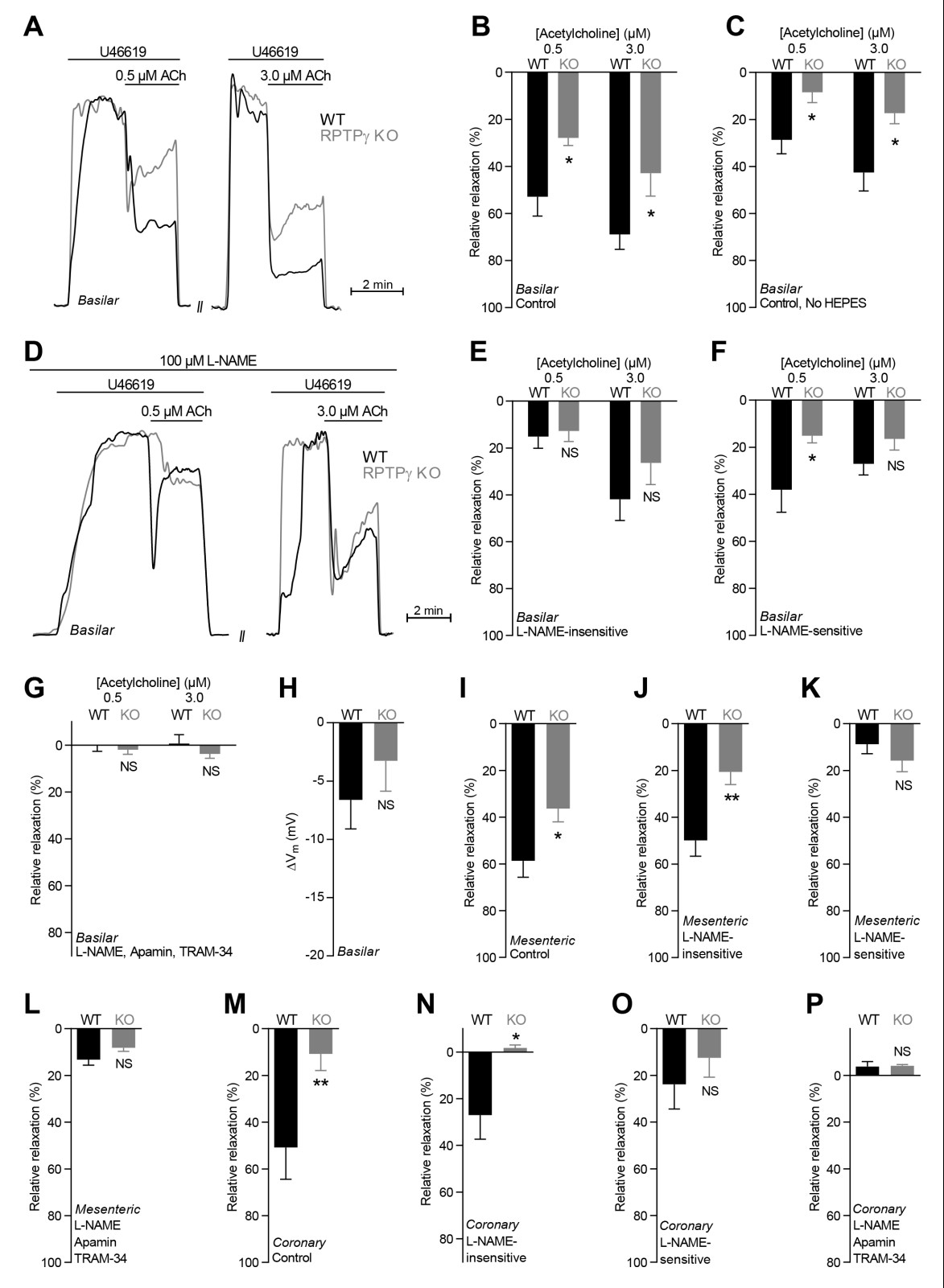

**Figure 2.** Acetylcholine (ACh)-induced endothelium-dependent vasorelaxation is compromised in basilar, mesenteric, and coronary arteries from RPTPγ KO mice when investigated in presence of $CO_2/HCO_3^-$. (**A** and **D**) Force recordings illustrating acetylcholine-induced vasorelaxation of U46619-contracted basilar arteries under control conditions (**A**) and after treatment with 100 μM L-NAME (**D**). The pre-contraction force development in response to U46619 was normalized in order to display relative relaxations. (**B**, **I**, and **M**) Acetylcholine-induced relaxations of basilar (**B**, n = 7–9),

Figure 2 continued

mesenteric (I, n = 11–13), and coronary (M, n = 7) arteries from RPTPγ KO and WT mice under control conditions, i.e. without inhibitors of endothelial function. For mesenteric and coronary arteries, we applied a concentration of 0.5 and 3.0 µM acetylcholine, respectively. (C) Experiment similar to that in panel B but performed in a $CO_2/HCO_3^-$-containing solution without HEPES (n = 9–10). (E, J, and N) Acetylcholine-induced relaxations of basilar (E, n = 7–9), mesenteric (J, n = 11–13), and coronary (N, n = 7) arteries from RPTPγ KO and WT mice after treatment with 100 µM L-NAME. For mesenteric and coronary arteries, we applied a concentration of 0.5 and 3.0 µM acetylcholine, respectively. (F, K, and O) Acetylcholine-induced L-NAME-sensitive relaxations of basilar (F, n = 7–9), mesenteric (K, n = 11–13), and coronary (O, n = 7) arteries from RPTPγ KO and WT mice calculated as the difference in relaxations with and without 100 µM L-NAME. For mesenteric and coronary arteries, we applied a concentration of 0.5 and 3.0 µM acetylcholine, respectively. (G, L, and P) Acetylcholine-induced relaxations of basilar (G, n = 7–9), mesenteric (L, n = 11–13), and coronary (P, n = 7) arteries from RPTPγ KO and WT mice after treatment with 100 µM L-NAME, 50 nM apamin, and 1.0 µM TRAM-34. For mesenteric and coronary arteries, we applied a concentration of 0.5 and 3.0 µM acetylcholine, respectively. (H) Vascular smooth muscle cell membrane potential responses ($\Delta V_m$) to 3.0 µM acetylcholine in basilar arteries from RPTPγ KO and WT mice in presence of 100 µM L-NAME. Data were compared using two-way ANOVA followed by Sidak's post-test (panel B, C, and E-G), non-parametric Mann-Whitney test (panel H, K-M, and O), unpaired two-tailed Student's t-test (panel I and J) or unpaired two-tailed t-test with Welch's correction (panel N and P). Comparisons in panel B and J were made after square root transformation. *p<0.05, **p<0.01, NS: not significantly different vs WT.

vasorelaxation (*Figures 2* and *3*) as these responses rely on the $Ca^{2+}$-sensitive NO-synthase and on intermediate- ($IK_{Ca}$) and small-conductance ($SK_{Ca}$) $Ca^{2+}$-activated $K^+$-channels, respectively (*Figure 5O*).

Intracellular $H^+$ and $Ca^{2+}$ can compete for buffer binding (*Batlle et al., 1993*), many $Ca^{2+}$ handling proteins are pH-sensitive (*Boedtkjer and Aalkjaer, 2012*), and pH is an important regulator of NO-synthase activity (*Boedtkjer et al., 2011*). Therefore, we next evaluated whether RPTPγ KO influences intracellular pH—measured using the pH-sensitive fluorophore BCECF—in endothelial cells of small arteries. In basilar (*Figure 5P*) as well as mesenteric (*Figure 5Q*) arteries, intracellular pH of the endothelial cells was similar for RPTPγ KO and WT mice, irrespective of whether experiments were performed with or without $CO_2/HCO_3^-$. Omission of $CO_2/HCO_3^-$ from the bath solutions had no net effect on endothelial steady-state intracellular pH (*Figure 5P,Q*), which is consistent with previous studies based on arteries from C57BL/6 mice (*Boedtkjer et al., 2011*; *Boedtkjer et al., 2012*).

## RPTPγ amplifies hyperventilation-induced blood pressure elevations

If the vasomotor impact of RPTPγ is sufficiently widespread to affect total peripheral resistance, we predicted that KO of RPTPγ would influence blood pressure regulation. We therefore measured systemic blood pressure and heart rate by radiotelemetry (*Figure 6A,B*). Based on these recordings, we observed no difference in resting blood pressure or heart rate (*Figure 6A*) or in the blood pressure-elevating effect of L-NAME (*Figure 6B*) between RPTPγ KO and WT mice.

As the lack of difference in resting blood pressure between WT and RPTPγ mice could be due to compensatory adaptations, we next studied acute blood pressure responses of endotracheally intubated, mechanically ventilated mice under capnographic control. When mice were ventilated to a normal expiratory end-tidal CO2 fraction ($F_{ET}CO_2$) of 3.5%, arterial blood gas parameters were very similar in RPTPγ KO and WT mice (*Table 1*) consistent with previous findings (*Zhou et al., 2016*). Hyperventilating mice until $F_{ET}CO_2$ was lowered to 2% caused the expected decrease in $P_aCO_2$ and increase in $pH_a$ (*Table 1*). The shift of the chemical equilibrium $CO_2 + H_2O \rightleftarrows HCO_3^- + H^+$ also resulted in the anticipated decrease in $[HCO_3^-]_a$ (*Table 1*). The mean arterial blood pressure of WT mice increased around 20% during hyperventilation and this blood pressure response was reduced by approximately 1/3 in RPTPγ KO mice (*Figure 6C*). Although the observed hyperventilation-induced reduction in arterial $[HCO_3^-]$ was of modest magnitude (1.4 mM, *Table 1*), the associated increase in blood pressure is consistent with attenuated RPTPγ-dependent vasodilator influences when extracellular $[HCO_3^-]$ decreases (*Boedtkjer et al., 2016c*). The remaining hyperventilation-induced blood pressure elevation likely results from a direct vasocontractile effect of alkalosis (*Boedtkjer et al., 2016c*).

## RPTPγ regulates cerebral perfusion

Metabolic waste products—that are generated at elevated rate and locally accumulate when neuronal activity increases—influence cerebral blood flow. We studied cerebral perfusion using laser

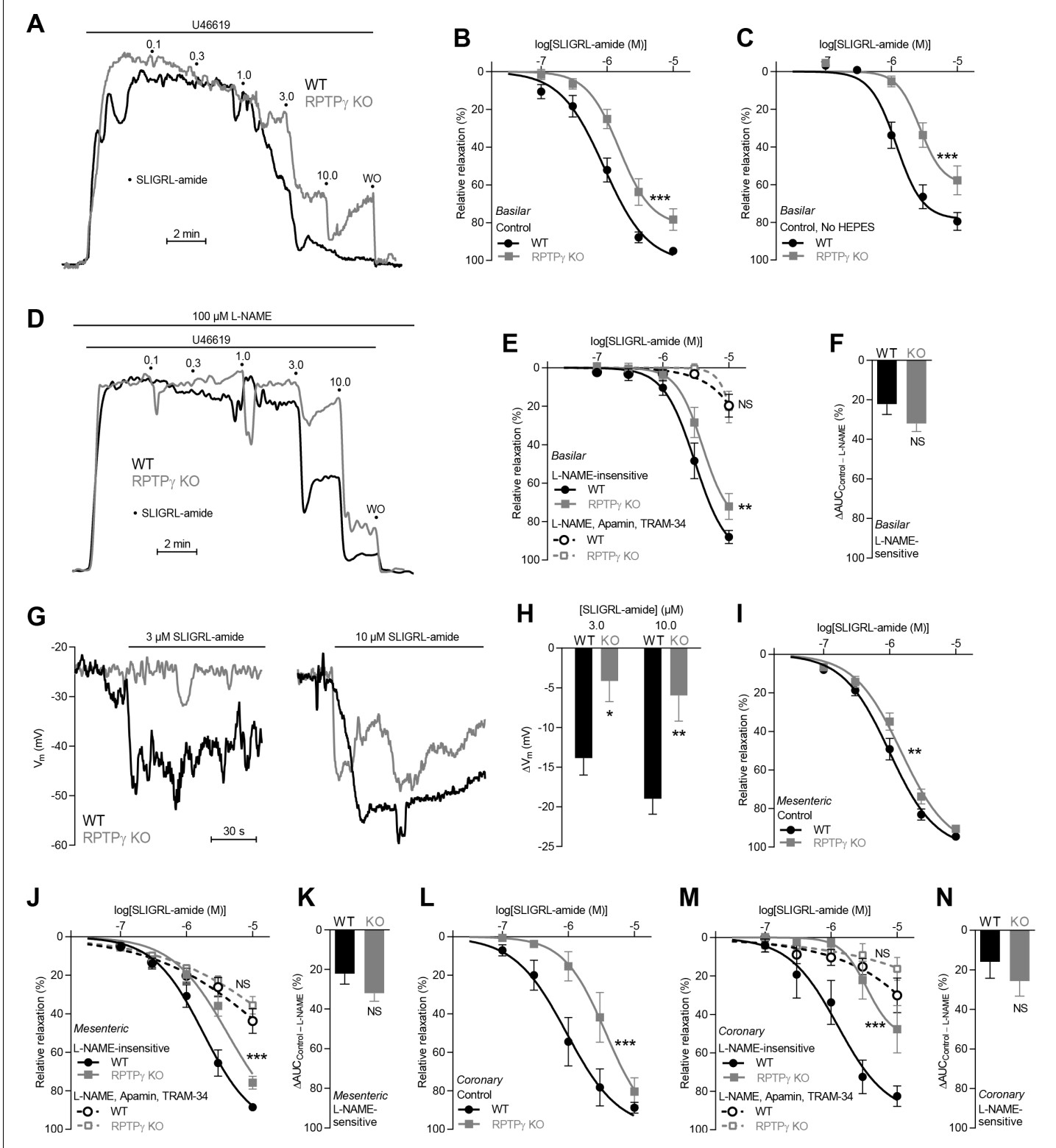

**Figure 3.** SLIGRL-amide-induced endothelium-dependent vasorelaxation is compromised in basilar, mesenteric, and coronary arteries from RPTPγ KO mice when investigated in presence of $CO_2/HCO_3^-$. (**A** and **D**) Force recordings illustrating SLIGRL-amide-induced vasorelaxation of U46619-contracted basilar arteries under control conditions (**A**) and after treatment with 100 µM L-NAME (**D**). The numbers indicate points of cumulative stepwise increases in SLIGRL-amide from 0.1 µM to 10.0 µM. WO indicates the point of washout. The pre-contraction force development in response to U46619 was normalized in order to display relative relaxations. (**B**, **I**, and **L**) SLIGRL-amide-induced relaxations of basilar (**B**, n = 12), mesenteric (**I**, n = 11), and

*Figure 3 continued on next page*

Figure 3 continued

coronary (L, n = 7) arteries from RPTPγ KO and WT mice under control conditions, that is without inhibitors of endothelial function. (C) Experiment similar to that in panel B but performed in a $CO_2/HCO_3^-$-containing solution without HEPES (n = 8). (E, J, M) SLIGRL-amide-induced relaxations of basilar (E, n = 12), mesenteric (J, n = 11), and coronary (M, n = 7) arteries from WT and RPTPγ KO mice after treatment with 100 μM L-NAME alone or in combination with 50 nM apamin and 1.0 μM TRAM-34. (F, K, and N) SLIGRL-amide-induced L-NAME-sensitive relaxations of basilar (F, n = 12), mesenteric (K, n = 11), and coronary (N, n = 7) arteries from RPTPγ KO and WT mice calculated as the difference in area under the curve (AUC) between the concentration-response curves with and without L-NAME. (G and H) Original recordings (G) and summarized data (H, n = 8–9) from basilar arteries showing vascular smooth muscle cell membrane potential ($V_m$) responses to SLIGRL-amide in presence of 100 μM L-NAME. In panel B, C, E, I, J, L, and M, the relative relaxations were fitted to sigmoidal curves and compared using extra sum-of-squares $F$ tests. In panel F, K, and N, the relative relaxations were compared by unpaired two-tailed Student's $t$-test. In panel H, membrane potential responses were compared by two-way ANOVA followed by Sidak's post-test. *$p<0.05$, **$p<0.01$, ***$p<0.001$. NS: not significantly different vs WT.

speckle imaging (*Figure 6D–F*) in order to evaluate the involvement of RPTPγ in sensing of acid-base disturbances. Tissue perfusion in the somatosensory barrel cortex increased twice as much in RPTPγ KO mice compared to WT mice during whisker stimulation (*Figure 6E*). This finding is consistent with metabolically produced $H^+$ leading to a local decrease in $[HCO_3^-]$ that is sensed via RPTPγ in arteries from WT mice (*Figure 5O*). As RPTPγ-dependent vasorelaxation wanes at low $[HCO_3^-]$ (*Boedtkjer et al., 2016c*), our data support that RPTPγ limits increases in cerebral perfusion induced by metabolic acidosis (*Figure 6E*).

Acid-base conditions during increased neuronal activity are complicated, as the increased metabolism will also induce some degree of $CO_2$ accumulation that tends to increase $[HCO_3^-]$. In order to produce a simpler acid-base disturbance with predictable changes in extracellular $[HCO_3^-]$, we next hyperventilated mice, which reduced $pCO_2$, increased pH, and lowered $[HCO_3^-]$ (*Table 1*). Under these conditions, cerebral vasoconstriction is expected in WT mice based on the combined effect of alkalosis and decreased $[HCO_3^-]$ (*Boedtkjer et al., 2016c*). If RPTPγ is required for sensing of $HCO_3^-$, only the effect of the elevated pH is expected in the RPTPγ KO mice; and indeed, we saw a 30% smaller drop in cerebral perfusion in RPTPγ KO mice compared to WT mice during hyperventilation (*Figure 6F*). These recordings may in fact underestimate the difference in cerebrovascular resistance between RPTPγ KO and WT mice since the cerebral perfusion pressure—based on the blood pressure changes shown in (*Figure 6C*)—increased less in hyperventilated RPTPγ KO than WT mice. When whisker stimulation was performed during hyperventilation, we saw an even more prominent difference in cerebral perfusion between WT and RPTPγ KO mice (*Figure 6F*).

Together, our findings demonstrate that RPTPγ plays a substantial role for control of blood pressure (*Figure 6C*) and cerebral perfusion (*Figure 6E,F*) during acid-base deviations where $[HCO_3^-]$ decreases below the normal level.

## *PTPRG* is a susceptibility locus for ischemic vascular disease

Based on human exome sequencing data from the UK Biobank, we next explored whether the functional role of RPTPγ in regulating endothelial function and responding to metabolic disturbances in murine cerebral, mesenteric, and coronary arteries is corroborated by and translates to an altered risk for human ischemic vascular disease amongst carriers of predicted loss-of-function variants within *PTPRG*. We identified 334 missense variants and 75 predicted loss-of-function variants (categorized as low, moderate, and high impact) in *PTPRG* among the ~50,000 UK Biobank participants with available exome sequencing data.

The 72 carriers of predicted loss-of-function *PTPRG* variants with moderate or high impact showed substantially elevated risk (*Figure 7A*) of a combined vascular disease phenotype encompassing the three vascular beds functionally investigated in this study (*Figures 2–6*): cerebral infarct (ICD-10 diagnosis code I63), angina pectoris (ICD-10 diagnosis code I20), acute myocardial infarction (ICD-10 diagnosis code I21), and acute vascular disorders of the intestine (ICD-10 diagnosis code K55.0). In contrast, the 123 carriers of predicted loss-of-function *PTPRG* variants with low impact as well as the 29,975 carriers of one or more missense *PTPRG* variants had a risk similar to non-carriers for this aggregated phenotype of acute vascular disease (*Figure 7A*).

We followed-up by investigating the separate occurrences of diagnosed angina pectoris, acute myocardial infarction, and cerebral infarction amongst the individuals, who carried a predicted loss-of-function variant with moderate or high impact. The carriers of these most severe loss-of-function

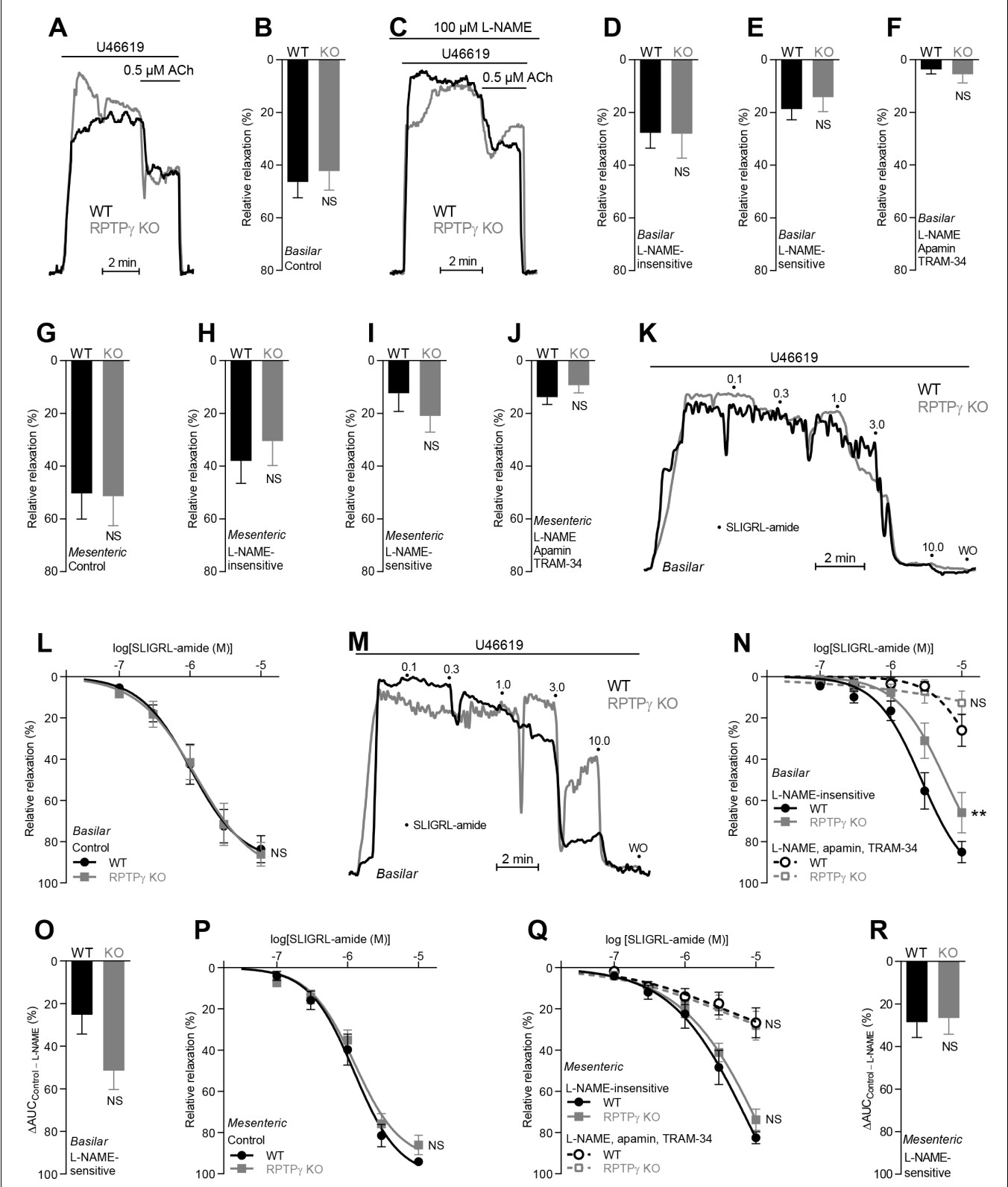

**Figure 4.** In absence of $CO_2/HCO_3^-$, acetylcholine (ACh)- and SLIGRL-amide-induced endothelium-dependent vasorelaxation of basilar and mesenteric arteries is similar between RPTPγ KO and WT mice. (A and C) Force recordings illustrating acetylcholine-induced vasorelaxation of U46619-contracted basilar arteries under control conditions (A) and after treatment with 100 µM L-NAME (C). The pre-contraction force development in response to U46619 was normalized in order to display relative relaxations. (B and G) Acetylcholine-induced relaxations of basilar (B, n = 11) and mesenteric (G,

*Figure 4 continued on next page*

Figure 4 continued

n = 6–7) arteries from RPTPγ KO and WT mice under control $CO_2/HCO_3^-$-free conditions, that is, without inhibitors of endothelial function. (**D** and **H**) Acetylcholine-induced relaxations of basilar (**D**, n = 11) and mesenteric (**H**, n = 6–7) arteries from RPTPγ KO and WT mice in presence of 100 μM L-NAME under $CO_2/HCO_3^-$-free conditions. (**E** and **I**) Acetylcholine-induced L-NAME-sensitive relaxations of basilar (**E**, n = 11) and mesenteric (**I**, n = 6–7) arteries from RPTPγ KO and WT mice calculated as the difference between relaxations with and without L-NAME. Experiments were performed in absence of $CO_2/HCO_3^-$. (**F** and **J**) Acetylcholine-induced relaxations of basilar (**F**, n = 11) and mesenteric (**J**, n = 6–7) arteries from RPTPγ KO and WT mice after treatment with 100 μM L-NAME, 50 nM apamin, and 1.0 μM TRAM-34 under $CO_2/HCO_3^-$-free conditions. (**K** and **M**) Force recordings illustrating SLIGRL-amide-induced vasorelaxation of U46619-contracted basilar arteries under control conditions (**K**) and after treatment with 100 μM L-NAME (**M**). The numbers indicate points of cumulative stepwise increases in SLIGRL-amide from 0.1 μM to 10.0 μM. WO indicates the point of washout. The pre-contraction force development in response to U46619 was normalized in order to display relative relaxations. (**L** and **P**) SLIGRL-amide-induced relaxations of basilar (**L**, n = 10) and mesenteric (**P**, n = 6–7) arteries from RPTPγ KO and WT mice under control $CO_2/HCO_3^-$-free conditions, that is, without inhibitors of endothelial function. (**N** and **Q**) SLIGRL-amide-induced relaxations of basilar (**N**, n = 10) and mesenteric (**Q**, n = 6–7) arteries from RPTPγ KO and WT mice after treatment with 100 μM L-NAME alone or in combination with 50 nM apamin and 1.0 μM TRAM-34. The experiments were performed in absence of $CO_2/HCO_3^-$. (**O** and **R**) SLIGRL-amide-induced L-NAME-sensitive relaxations of basilar (**O**, n = 10) and mesenteric (**R**, n = 6–7) arteries from RPTPγ KO and WT mice calculated as the difference in area under the curve (AUC) between the concentration-response curves with and without L-NAME. In panels A-J, we applied a concentration of 0.5 μM acetylcholine. Data in panels B, E, G-J, and O were compared by unpaired two-tailed Student's $t$-tests, data in panels D and R by non-parametric Mann-Whitney test, and data in panel F by unpaired two-tailed $t$-test with Welch's correction. In panels L, N, P, and Q, the relative relaxations were fitted to sigmoidal curves and compared using extra sum-of-squares $F$ tests. Data in panel J were log-transformed before comparison. **$p<0.01$, NS: not significantly different vs WT.

variants had dramatically increased risk (~7 fold) of cerebral infarction whereas the risk of acute myocardial infarction showed a tendency toward elevation that did not reach statistical significance (*Figure 7B*). The number of exome-sequenced UK Biobank participants with acute intestinal vascular disease was too low to provide statistical power for a separate analysis.

To complement the evidence based on ICD-10 diagnosis codes, we further evaluated whether carriers of missense and loss-of-function variants in *PTPRG* differed in self-reported diseases compared to non-carriers (*Figure 7C*). The risk of self-reported heart attacks was greatly elevated (~4 fold) amongst individuals carrying loss-of-function *PTPRG* variants of moderate and high impact (*Figure 7C*). The risk of self-reported stroke also showed a strong tendency toward elevation—with borderline statistical significance—amongst the carriers of moderate- and high-impact loss-of-function *PTPRG* variants (*Figure 7C*).

As cardiac pumping function often deteriorates following coronary ischemia, we next evaluated the cardiac contractile function in *PTPRG* loss-of-function carriers amongst the 12,851 exome-sequenced UK Biobank participants with available cardiac magnetic resonance imaging (MRI) data. We observed a significantly lower left ventricular ejection fraction in carriers of high-impact loss-of-function *PTPRG* variants compared to non-carriers (*Figure 7D*) consistent with the greater risk of ischemic heart disease (*Figure 7C*).

The statistical analyses linking *PTPRG* to risk of ischemic vascular disease (*Figure 7A–C*) and reduced left ventricular ejection fraction (*Figure 7D*) were all adjusted for sex, age, body mass index, genetic principal component, smoking status, dyslipidemia, diabetes, and hypertension between carriers and non-carriers of the evaluated *PTPRG* variants.

Taken together, we demonstrate that RPTPγ (a) $CO_2/HCO_3^-$-dependently enhances endothelial intracellular $Ca^{2+}$ responses and endothelium-dependent vasorelaxation, (b) regulates microvascular perfusion and blood pressure during acid-base disturbances, and (c) is associated with human ischemic vascular disease of the brain and heart.

## Discussion

Local acid-base-dependent mechanisms of arterial tone regulation match tissue perfusion to the oxidative metabolic demand and thereby dynamically control cardiovascular function during cycles of intermittent rest and activity. Re-establishing proper tissue perfusion is of obvious clinical importance when managing patients with latent or fulminant ischemia. Responses of coronary and cerebral arteries to acid-base disturbances have been recognized for almost 140 years (*Gaskell, 1880*; *Lassen, 1968*) but this information has not yet been harnessed for therapeutic intervention for lack of identified molecular players and their interactions. In the current study, we identify *PTPRG* as a susceptibility locus for human ischemic vascular disease (*Figure 7*) and provide mechanistic evidence

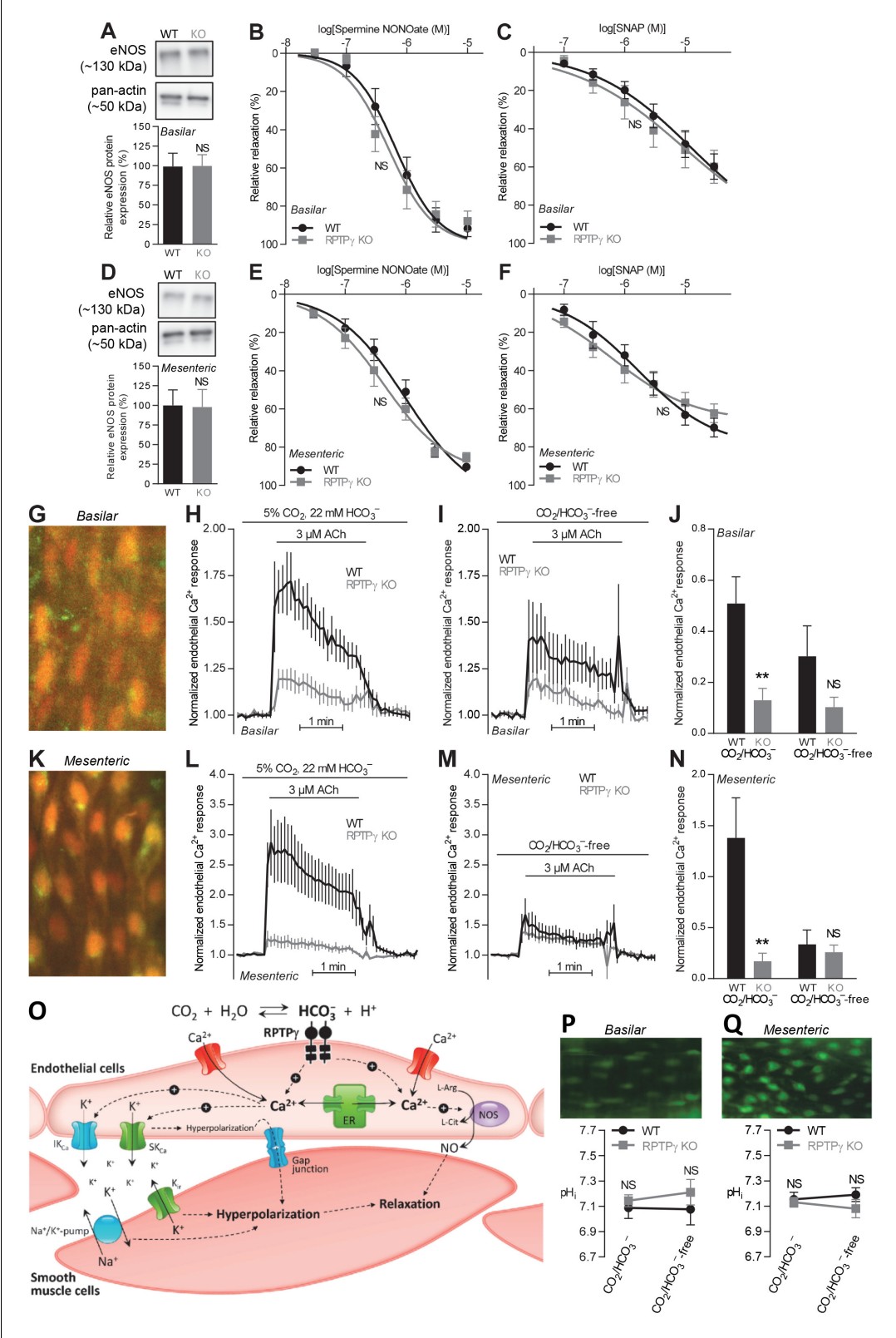

**Figure 5.** Acetylcholine (ACh)-induced endothelial intracellular $Ca^{2+}$ responses are lower in arteries from RPTPγ KO than WT mice. In contrast, expression of endothelial NO-synthase (eNOS), responses to NO-donors, and endothelial intracellular pH are unaffected in basilar and mesenteric arteries from RPTPγ KO compared to WT mice. (**A** and **D**). Expression of eNOS in basilar (**A**) and mesenteric (**D**) arteries from RPTPγ KO and WT mice (n = 5–6). Representative bands are shown next to results from densitometric analyses where the expression levels relative to pan-actin are normalized

*Figure 5 continued on next page*

Figure 5 continued

to the average level in arteries from WT mice. (**B** and **E**) Spermine NONOate-induced relaxations of basilar (**B**, n = 10–14) and mesenteric (**E**, n = 9) arteries from RPTPγ KO and WT mice. (**C** and **F**) SNAP-induced relaxations of basilar (**C**, n = 10–15) and mesenteric (**F**, n = 9) arteries from RPTPγ KO and WT mice. All experiments in panels **B**, **C**, **E**, and **F** were performed on arteries pre-contracted with U46619, in presence of $CO_2/HCO_3^-$, and after treatment with 100 μM L-NAME, 50 nM apamin, and 1.0 μM TRAM-34. (**G** and **K**) Overlaid fluorescence images of endothelial cells in basilar (**G**) and mesenteric (**K**) arteries loaded by luminal perfusion with Fluo-4 (green) and Fura Red (red). (**H**, **I**, **L**, and **M**) Average traces of acetylcholine-induced endothelial intracellular $Ca^{2+}$ responses in basilar (**H** and **I**) and mesenteric (**L** and **M**) arteries from RPTPγ KO and WT mice (n = 6–9) in presence (**H** and **L**) and absence (**I** and **M**) of $CO_2/HCO_3^-$. The evaluation of endothelial $Ca^{2+}$ responses was based on the $F_{505-530}/F_{>600}$ ratio normalized to the average ratio during the first two minutes of the individual recordings. (**J** and **N**) Summarized endothelial intracellular $Ca^{2+}$ responses in basilar (**J**) and mesenteric (**N**) arteries upon acetylcholine stimulation (n = 6–9). The endothelial $Ca^{2+}$ response was calculated as the average increase in normalized fluorescence during the 2 min acetylcholine exposure compared to the baseline value before activation. (**O**) Schematic illustration of how RPTPγ influences vasomotor functions of resistance arteries. (**P** and **Q**) Steady-state intracellular pH of endothelial cells in basilar (**P**) and mesenteric (**Q**) arteries from RPTPγ KO and WT mice (n = 5–7) in the presence and absence of $CO_2/HCO_3^-$. In panels **A** and **D**, expression levels were compared with unpaired two-tailed Student's t-tests. In panels **B**, **C**, **E**, and **F**, the relative relaxations were fitted to sigmoidal curves and compared using extra sum-of-squares F tests. In panels **J** and **N**, data were compared by two-way ANOVA followed by Sidak's post-tests. In panels **P** and **Q**, data were compared by repeated-measures two-way ANOVA followed by Sidak's post-tests. The data in panel **N** were log-transformed before comparisons. \*p<0.05, \*\*p<0.01, \*\*\*p<0.001, NS: not significantly different vs WT. ER, endoplasmic reticulum; $K_{ir}$, inward-rectifier $K^+$-channel; $IK_{Ca}$, intermediate-conductance $K^+$-channel; NOS, NO-synthase; $SK_{Ca}$, small-conductance $K^+$-channel; SNAP, S-nitroso-N-acetylpenicillamine.

that RPTPγ regulates endothelial intracellular $Ca^{2+}$ responses (*Figure 5H,L*), endothelium-dependent vasorelaxation (*Figures 2* and *3*), cerebral perfusion (*Figure 6E,F*), and blood pressure (*Figure 6C*) during acid-base disturbances and increased metabolic demand.

Variation in buffer composition accompanies changes in pH during acid-base disturbances. We show that the $CO_2/HCO_3^-$ buffer plays a hitherto unappreciated role for regulation of endothelial function and arterial tone through mechanisms that require RPTPγ (*Figures 2–4*). As schematically illustrated in *Figure 5O*, we demonstrate that RPTPγ enhances endothelial intracellular $Ca^{2+}$ signals (*Figure 5G–N*) that in turn activate endothelial NO synthesis and EDH-type responses (*Figures 2* and *3*). These effects are in congruence with the $Ca^{2+}$-sensitivity of eNOS and the SK and IK $Ca^{2+}$-activated $K^+$-channels. The vasorelaxant influence of RPTPγ wanes at low extracellular $[HCO_3^-]$ (*Boedtkjer et al., 2016c*), which attenuates by half the elevation of cerebral perfusion during sensory input (*Figure 6E*) and amplifies by 30% the decrease in perfusion during hyperventilation (*Figure 6F*). We also observe that RPTPγ is necessary for approximately 1/3 of the blood pressure increase during hyperventilation (*Table 1* and *Figure 6C*). Considering the prominent influence of RPTPγ on endothelial function in several vascular beds, somewhat surprisingly, we observe blood pressure consequences of RPTPγ KO only when we impose acute acid-base disturbances (*Figure 6A–C*). The unaltered resting blood pressure in RPTPγ KO mice (*Figure 6A*) most likely reflects that numerous discrete mechanisms are involved in blood pressure control and that compensation—for example, through nervous, hormonal or renal influences (*Boedtkjer and Aalkjaer, 2013b*)—can maintain blood pressure in the sustained phase and mask hemodynamic consequences of RPTPγ except when acute acid-base disturbances are imposed.

Previously recognized cellular acid-base sensors include G-protein coupled receptors (e.g. OGR1, GPR4, and TDAG8), ion channels (e.g. ASIC, $BK_{Ca}$), and enzymes (e.g. NO-synthase and rho-kinase) sensitive to $H^+$ (*Boedtkjer et al., 2011*; *Boedtkjer et al., 2012*; *Fleming et al., 1994*; *Schubert et al., 2001*; *Ludwig et al., 2003*; *Wemmie et al., 2006*; *Wenzel et al., 2020*). In addition, we and others have described cellular functions responsive to changes in $[HCO_3^-]$ and modified by RPTPγ (*Zhou et al., 2016*; *Boedtkjer et al., 2016c*) or the soluble adenylyl cyclase (*Chen et al., 2000*). Evidence connecting acid-base sensors to human disease has so far been scarce; but in the current study, we identify *PTPRG* as an ischemia susceptibility locus (*Figure 7*), which is supported by a recent meta-analysis of genome-wide association studies linking *PTPRG* to ischemic stroke in African Americans (*Carty et al., 2015*).

Using out-of-equilibrium technology—that permits separate control of pH and the individual $CO_2/HCO_3^-$ buffer components—we previously demonstrated that RPTPγ responds to changes in extracellular $[HCO_3^-]$ independently of $pCO_2$ and pH (*Boedtkjer et al., 2016c*). The homology between the extracellular domain of RPTPγ and the active site of the carbonic anhydrases supports the ability of RPTPγ to bind $HCO_3^-$ even though it lacks histidine residues required for carbonic anhydrase activity (*Zhou et al., 2016*). The intracellular aspect of RPTPγ contains phosphatase domains

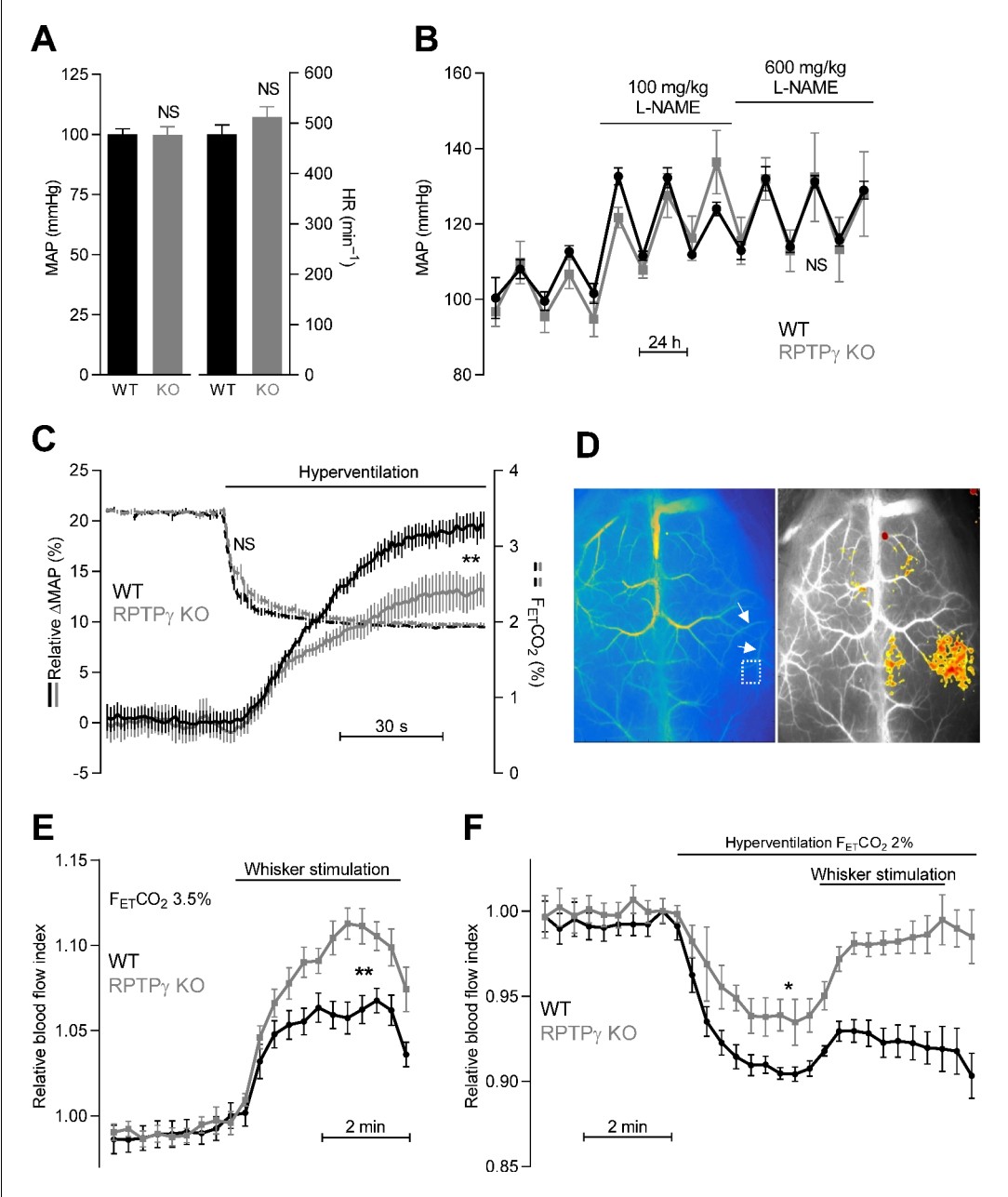

**Figure 6.** Systemic blood pressure and heart rate are similar in WT and RPTPγ KO mice at rest. In contrast, RPTPγ KO mice show elevated cerebrovascular perfusion during whisker stimulation, attenuated hyperventilation-induced blood pressure elevation, and diminished hyperventilation-induced reduction in cerebrovascular perfusion. (A) Resting mean arterial pressure (MAP) and heart rate (HR) measured by radiotelemetry in WT and RPTPγ KO mice (n = 6–7). Data were compared by unpaired two-tailed Student's t-tests. (B) Mean arterial pressure under control conditions and during intake of 100 and 600 mg/kg/day L-NAME in the drinking water (n = 4–5). Data were square root-transformed and then compared by repeated-measures two-way ANOVA. (C) Changes in relative mean arterial blood pressure (ΔMAP, solid lines) and expiratory end-tidal $CO_2$ fraction ($F_{ET}CO_2$, dashed lines) during hyperventilation (n = 13). The steady-state values averaged over the last 30 s of the stimulation period were compared between WT and RPTPγ KO mice by unpaired two-tailed Student's t-test (ΔMAP) or non-parametric Mann-Whitney test ($F_{ET}CO_2$). In the anesthetized mice, the absolute mean arterial blood pressure prior to hyperventilation was 63 ± 5 mmHg in RPTPγ KO compared to 60 ± 3 mmHg in WT mice (p=0.63, unpaired two-tailed Student's t-test). (D) Exemplar laser speckle images. The left image shows the second posterior bifurcation of the middle cerebral artery (arrows) that was used to locate the same region of interest (dotted square) in each experiment. The right image provides a 7.5% threshold response map from a whisker stimulation experiment in an RPTPγ KO mouse. (E and F) Relative cerebrovascular perfusion in the barrel cortex (dotted square in panel D) of RPTPγ KO and WT mice during whisker stimulation (E, n = 9) and during hyperventilation alone or combined with whisker stimulation (F, n = 8). The steady-state blood flow index averaged over the last minute of the intervention period was compared between WT and

*Figure 6 continued on next page*

*Figure 6 continued*

RPTPγ KO mice by unpaired two-tailed Student's *t*-test (whisker stimulation) or unpaired two-tailed *t*-test with Welch's correction (hyperventilation). *p<0.05, **p<0.01, NS: not significantly different vs WT. $F_{ET}CO_2$ indicates the expiratory end-tidal $CO_2$ fraction measured by capnography.

with suggested auto-inhibitory activity; and $HCO_3^-$ possibly alters the phosphatase activity by influencing the degree of RPTPγ dimerization (*Barr et al., 2009*). The separate signaling effects of $H^+$ and $HCO_3^-$ are further confirmed by the similar steady-state intracellular pH of endothelial cells in arteries from WT and RPTPγ KO mice (*Figure 5P,Q*).

The current study adds to our growing appreciation that acid-base equivalents, and $HCO_3^-$ in particular, fulfill multifaceted functions (*Boedtkjer et al., 2016a*). In cerebral resistance arteries, $HCO_3^-$ (a) contributes to buffering of acute acid loads (*Rasmussen and Boedtkjer, 2018*), (b) serves as substrate particularly for the $Na^+,HCO_3^-$-cotransporter NBCn1 that protects against intracellular acidification (*Thomsen et al., 2014*), and (c) is sensed by RPTPγ to regulate cerebral perfusion (*Figure 6*). In migrating vascular smooth muscle cells from conduit arteries, the high spatial mobility of the $CO_2/HCO_3^-$ buffer system also contributes to dissipating local pH gradients in diffusion-restricted spaces of filopodia (*Boedtkjer et al., 2016b*).

Our current (*Figures 1–4* and *6*) and previous (*Boedtkjer et al., 2016c*) findings demonstrate that the endothelium of resistance arteries senses the local acid-base composition and has capacity to modify vascular resistance and perfusion in response to disturbances in extracellular $[HCO_3^-]$. Earlier studies of mouse arteries provide additional evidence that intracellular acidification inhibits endothelial NO-synthesis (*Boedtkjer et al., 2011*; *Boedtkjer et al., 2012*; *Thomsen et al., 2014*), intracellular alkalinisation interferes with myo-endothelial current transfer required for EDH-type responses (*Boedtkjer et al., 2013a*), and acid-base disturbances modify prostanoid-mediated endothelium-dependent vasocontraction (*Baretella et al., 2014*). Together, these studies highlight the sophisticated nature of vasomotor control that integrate the requirement for local blood flow and the necessity to minimize deviations in capillary pressure and fluid filtration during local metabolic disturbances, perturbed nutrient delivery, and restricted waste product elimination. Acting as a brake on vasodilation in regions of unmet metabolic demand, RPTPγ could reduce the degree of edema and consequent tissue damage caused by unopposed $H^+$-induced vasodilation (*Boedtkjer, 2018*; *Boedtkjer et al., 2016a*).

As illustrated in *Figure 1D and E*, *Ptprg* mRNA is found in arteriolar, capillary, as well as venular endothelial cells (*Kalucka et al., 2020*; *Vanlandewijck et al., 2018*), and it remains a future task to

**Table 1.** Arterial blood gas measurements from RPTPγ KO and WT mice (n = 7–9) under control conditions (normoventilation, $F_{ET}CO_2$ = 3.5%) and after 2 min of hyperventilation ($F_{ET}CO_2$ = 2%).
We compared data by repeated-measures two-way ANOVA. $F_{ET}CO_2$, expiratory end-tidal $CO_2$ fraction.

| Arterial blood gasses | WT | | RPTPγ KO |
|---|---|---|---|
| *Control*, $F_{ET}CO_2$ = 3.5% | | | |
| $pH_a$ | 7.28 ± 0.01 | | 7.28 ± 0.01 |
| $P_aCO_2$ (mmHg) | 45.2 ± 1.2 | | 46.2 ± 1.8 |
| $[HCO_3^-]_a$ (mM) | 20.4 ± 0.6 | | 20.9 ± 0.8 |
| *Hyperventilation*, $F_{ET}CO_2$ = 2% | | | |
| $pH_a$ | 7.38 ± 0.02 | | 7.41 ± 0.01 |
| $P_aCO_2$ (mmHg) | 32.3 ± 0.9 | | 32.3 ± 1.5 |
| $[HCO_3^-]_a$ (mM) | 19.0 ± 0.9 | | 19.2 ± 0.8 |
| Statistics (repeated-measures two-way ANOVA) | p-Values | | |
| | Genotype | Ventilation | Interaction |
| $pH_a$ | 0.40 | <0.001 | 0.19 |
| $P_aCO_2$ | 0.90 | <0.001 | 0.06 |
| $[HCO_3^-]_a$ | 0.69 | <0.001 | 0.47 |

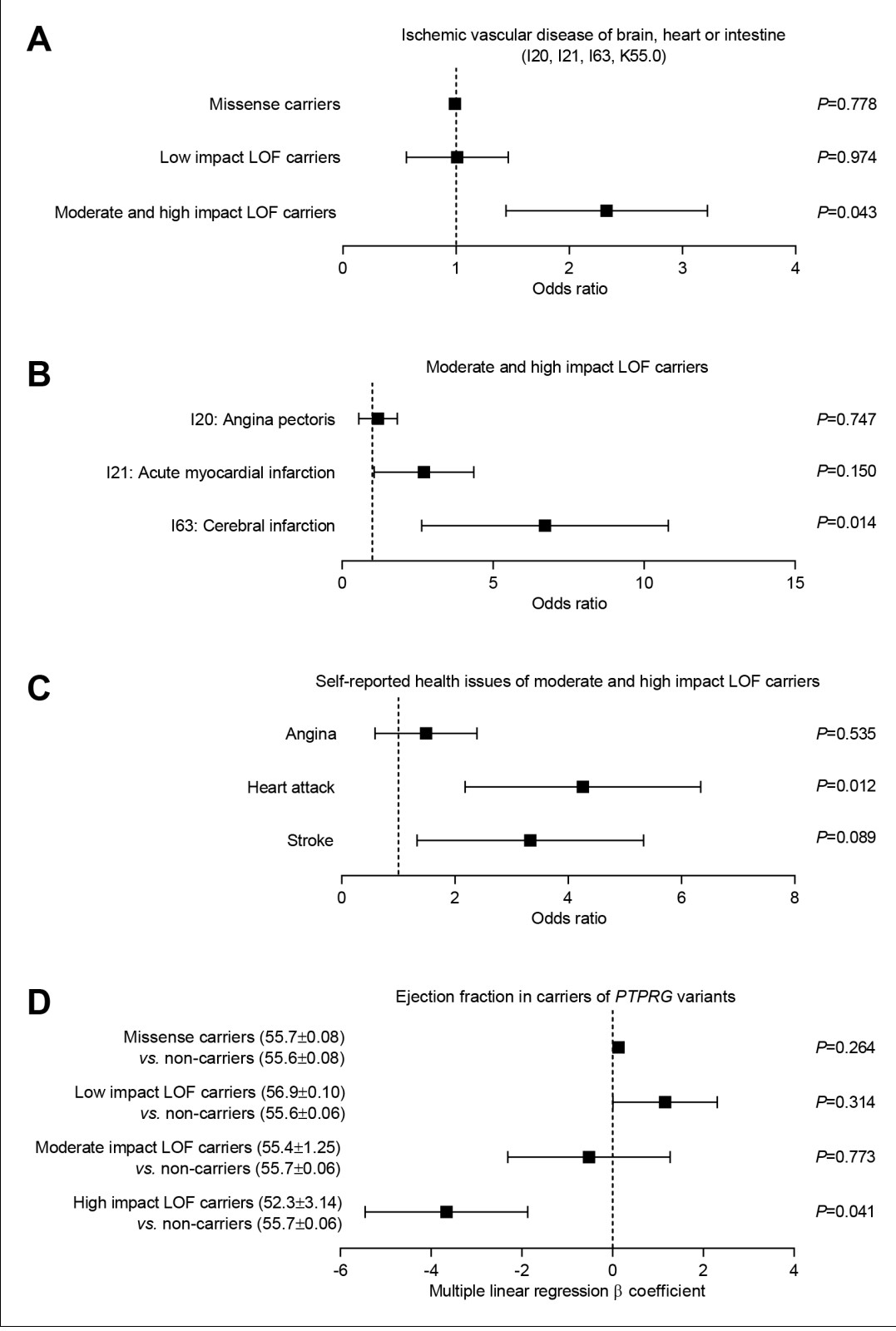

**Figure 7.** Burden analysis of loss-of-function variants in *PTPRG* reveals association with human ischemic vascular disease. (**A**) Association of *PTPRG* missense variants (localized outside exon-intron boundary regions) and predicted loss-of-function (LOF) variants of low, moderate, and high impact with ischemic cerebral, coronary, and mesenteric vascular disease (aggregate of ICD-10 diagnosis codes I20, I21, I63, and K55.0). (**B** and **C**) Association

*Figure 7 continued*
of predicted moderate- and high-impact loss-of-function variants in *PTPRG* with diagnosed (**B**) and self-reported (**C**) coronary and cerebral ischemia. (**D**) Association between missense and loss-of-function variants in *PTPRG* and left ventricular ejection fraction. The p-values in panel **A** through **C** come from logistic regression analyses and the p-values in panel **D** from multiple linear regression analyses. In each case, we adjusted for sex, age, body mass index, genetic principal component, smoking status, dyslipidemia, diabetes, and hypertension.

evaluate physiological and pathophysiological roles of RPTPγ in these individual blood vessel segments. Transcript levels for *Ptprg* appear particularly high in a smaller subset of endothelial cells (up to 20–40%, depending on the vascular bed; *Figure 1C–E*), which raises the intriguing possibility that a distinct population of endothelial cells act as the primary sensors of the metabolic environment.

Hyperventilation-induced cerebral vasoconstriction can cause syncope but is also a therapeutic tool for rapidly reducing cerebral blood flow, capillary filtration, and intracranial pressure. Hypocapnia during hyperventilation leads to decreased [$HCO_3^-$] and alkalosis (*Table 1*); and the current study demonstrates that RPTPγ is necessary for approximately 30% of the associated decrease in cerebral perfusion (*Figure 6F*).

In conclusion, the single-pass transmembrane protein RPTPγ enhances endothelium-dependent vasorelaxation of resistance arteries—by amplifying endothelial intracellular $Ca^{2+}$ responses, NO synthesis, and EDH-type responses—and adjusts cerebral perfusion during increased neuronal activity and acid-base disturbances. Although RPTPγ shows no major impact on resting blood pressure, it contributes markedly to the blood pressure increase observed during hyperventilation. Supporting the translational value of our findings, predicted loss-of-function variants in *PTPRG* are associated with human ischemic vascular disease in the brain and heart. The influence of RPTPγ on vascular resistance specifically in tissues with disturbed acid-base composition makes RPTPγ a promising focus for targeted therapeutic intervention against ischemia.

## Materials and methods

RPTPγ KO mice (MGI:3663183) from Dr. Joseph Schlessinger, Yale University, USA (*Lamprianou et al., 2006*), were kindly provided by Dr. Walter F. Boron, Case Western Reserve University, USA (*Zhou et al., 2016*). The transgenic mice were originally generated by inserting an internal ribosome entry site–β-galactosidase–neomycin–poly(A) cassette immediately 3′ of exon 1 (*Lamprianou et al., 2006*). Homologous recombination in W4 129S6/SvEv mouse embryonic stem cells (RRID:CVCL_Y634) was achieved following electroporation, and correctly targeted embryonic stem cells were used for embryo aggregation to achieve germ line chimeras (*Lamprianou et al., 2006*). At the time of experiments, the transgene-carrying mice had been backcrossed to the C57BL/6 line for at least eight generations. In total, we used 125 WT and 112 RPTPγ KO mice that were kept on a 12 hr light/12 hr dark cycle with ad libitum access to tap water and chow (Altromin 1324; Brogaarden, Denmark). We used Eco-Pure Aspen Chips 2 Premium cage bedding (Datesand, UK). Genomic DNA used for genotyping was isolated from tail biopsies collected at weening and again after sacrifice. All animal experimental procedures were approved by the Danish Animal Experiments Inspectorate (2016-15-0201-00982).

### Small vessel myography

Male RPTPγ KO and matched WT mice (>8 weeks old) deeply anesthetized by intraperitoneal injection of pentobarbital were sacrificed by exsanguination. Approximately 2 mm long segments of basilar, mesenteric, and coronary septal arteries were dissected under a stereomicroscope, mounted in four-channel wire myographs (610M; DMT, Denmark) for isometric investigation, heated to 37°C, and normalized to 90% of the internal diameter corresponding to a transmural pressure of 100 mmHg (*Mulvany and Halpern, 1977*). The myograph chambers were aerated with 5% $CO_2$/balance air (for $CO_2$/$HCO_3^-$-containing physiological saline solutions) or nominally $CO_2$-free air (for $CO_2$/$HCO_3^-$-free solutions). The $CO_2$/$HCO_3^-$-containing solution consisted of (in mM): 119 NaCl, 22 NaHCO_3, 10 HEPES, 1.2 $MgSO_4$, 2.82 KCl, 5.5 glucose, 1.18 $KH_2PO_4$, 0.03 EDTA, 1.6 $CaCl_2$. The $CO_2$/$HCO_3^-$-free solution was produced by substituting NaHCO_3 with NaCl. HEPES is necessary for pH control in the $CO_2$/$HCO_3^-$-free solution; and we included HEPES also in the $CO_2$/$HCO_3^-$-containing

solution in order to separate vascular effects of $CO_2/HCO_3^-$ omission from potential effects of adding HEPES (*Altura et al., 1980*). For the control experiments in *Figures 2C* and *3C*, we used a $CO_2/HCO_3^-$-containing solution without HEPES, i.e., where the 10 mM HEPES was substituted with 5 mM NaCl. After all contents were added, the solutions were vigorously bobbled with the appropriate $CO_2$-containing or nominally $CO_2$-free gas mixture at 37°C before pH was adjusted to 7.40 while allowing ample time for buffer equilibration. Arterial force was acquired using a PowerLab 4/25 recorder and LabChart 7 Pro software (RRID:SCR_001620; ADInstruments, New Zealand). As standard warm-up protocol, arteries were exposed twice to 3 µM of the thromboxane $A_2$ analog U46619 for 2 min, and arteries that produced less than 1 mN force were excluded from further analysis. Vasorelaxation in response to acetylcholine, the PAR2 agonist SLIGRL-amide, and the NO-donors SNAP, and spermine NONOate was tested in U46619-contracted arteries that developed stable tension equivalent to ~70% of the maximal response during the standard warm-up procedure. Vasorelaxation, from the pre-contraction level, was quantified as the relative decrease in active tone during the last 30 s of each 2 min agonist application. We used two NO donors that differ, for instance, in their rate and location of NO release (*Bradley and Steinert, 2015*). Because acetylcholine-induced vasorelaxation shows substantial tachyphylaxis in arteries from C57BL/6 mice, we tested single doses of acetylcholine applied on separate U46619 pre-contractions. In contrast, SLIGRL-amide does not show this degree of tachyphylaxis, and we therefore tested its vasorelaxant properties through cumulative additions.

## Endothelial $Ca^{2+}$ measurements

$Ca^{2+}$-sensitive fluorophores were loaded preferentially into endothelial cells by slow (~1 mL/hr for 45 min) perfusion of physiological saline solution containing 2.5 µM Fluo-4 and 3.5 µM Fura Red (Invitrogen) through the lumen of basilar and mesenteric arteries mounted in a pressure myograph (120CP; DMT) (*Boedtkjer et al., 2011*). The cell permeant acetoxymethyl ester (AM) forms of the fluorophores were first solubilized in a load mix containing dimethyl sulfoxide, Pluronic F127, and Cremophor EL. The loaded arteries were next mounted in a confocal wire myograph (360CW; DMT) and studied using a Zeiss Axiovert 200M confocal microscope equipped with an LSM Pascal exciter and a 40 × objective (LD C-Apochromat; N.A. 1.10; Zeiss, Germany). The arteries were excited at 488 nm and emission light collected at wavelengths in the range of 505–530 nm ($F_{505-530}$; representing Fluo-4 signals that increase at elevated $[Ca^{2+}]$) and longer than 600 nm ($F_{>600}$; representing Fura Red signals that decrease at elevated $[Ca^{2+}]$). Arteries with no or only solitary loaded endothelial cells were excluded from analysis. The $F_{505-530}/F_{>600}$ ratio normalized to the baseline ratio (the first 2 min of the recording) was used to evaluate relative intracellular $Ca^{2+}$ dynamics.

## Endothelial intracellular pH measurements

Inverted basilar and mesenteric arteries were mounted on a 40 µm wire in a confocal wire myograph (360CW, DMT) and loaded with the pH-sensitive fluorophore 2′,7′-bis-(2-carboxyethyl)−5-(and-6)-carboxyfluorescein (BCECF) in physiological saline solution (1 µM BCECF-AM in 0.05‰ DMSO) to preferentially load endothelial cells, essentially as previously described (*Boedtkjer et al., 2011*; *Boedtkjer and Aalkjaer, 2009*). The loaded arteries were studied using an Olympus IX83 microscope equipped with a 20 × objective (LUCPlanFLN; N.A. 0.45; Olympus, Japan) and an ORCA-Flash 4.0 camera (Hamamatsu, Japan). The arteries were alternatingly excited at 490 and 436 nm and emission light collected at 530 nm. The $F_{490}/F_{436}$ fluorescence ratio was calibrated to intracellular pH using the high-$[K^+]$ nigericin method (*Boedtkjer and Aalkjaer, 2009*; *Aalkjaer and Cragoe, 1988*), which in arteries agrees well with the null-point technique (*Danielsen et al., 2013*). Steady-state intracellular pH was recorded in the presence and absence of $CO_2/HCO_3^-$.

## Membrane potential recordings

Membrane potentials were measured in vascular smooth muscle cells of isolated basilar arteries using sharp electrodes as previously described (*Boedtkjer et al., 2013a*). The arteries were mounted in a wire myograph (420A; DMT) and microelectrodes that had resistances of 40–120 MΩ when backfilled with 3 M KCl were inserted into the vascular wall from the adventitial side. Cell impalement was observed as a sudden drop in voltage followed by sharp return to baseline upon

retraction. Measurements with more than 10 mV difference between the baseline recording before and after impalement were excluded from analysis.

### *LacZ* reporter studies

The genetic insert that disrupts RPTPγ expression in the employed KO mice contains a promotorless *LacZ* sequence allowing for β-galactosidase expression under control of the *Ptprg* promotor (*Lamprianou et al., 2006*). Four homozygous KO mice (*Ptprg$^{LacZ/LacZ}$*) and two WT mice were perfusion fixed with 4% (weight/volume) paraformaldehyde in phosphate-buffered saline (PBS, in mM: 137 NaCl, 2.5 KCl, 4.3 Na$_2$HPO$_4$, and 1 KH$_2$PO$_4$) and investigated for promoter activity essentially as previously described (*Boedtkjer et al., 2008*). Segments of basilar, middle cerebral, gracilis, coronary, mesenteric, and pulmonary arteries as well as thoracic aorta were dissected free from surrounding tissue and washed in PBS overnight at 4℃. The arteries were then placed in staining solution (in mM: 5 K$_4$Fe(CN)$_6$, 5 K$_3$Fe(CN)$_6$, 2 MgCl$_2$, 0.1% (weight/volume) sodium dodecyl sulfate (SDS), 0.1% (volume/volume) TWEEN−20, and 0.1% (weight/volume) 5-bromo-4-chloro-indolyl-13-D-galactoside (X-Gal)) for 24 hr at room temperature (~21℃). Finally, samples were transferred to PBS containing 1% (weight/volume) ethylenediaminetetraacetic acid (EDTA) and 4% paraformaldehyde in order to stop the staining reaction. Whole mount micrographs of the arteries were captured using a Leica M165 C stereomicroscope equipped with a Leica M170 HD camera (Germany). In addition, we paraffin-embedded and cut basilar and middle cerebral arteries—that showed the strongest staining—to 4-μm-thick histological sections that were visualized on an upright Leica DM light microscope equipped with a Leica DM300 digital camera in order to identify the cellular expression pattern.

### Reverse transcription and quantitative polymerase chain reaction

Arteries dissected free from surrounding tissue in cold physiological saline solution were stored in RNALater (Qiagen, Denmark) at 4℃. The isolated arteries were homogenized in RLT lysis buffer (Qiagen) using a TissueLyser II (Qiagen) at 30 Hz for 2 min. RNA was isolated using the RNeasy Micro Qiacube kit including carrier RNA (Qiagen). Samples were reverse transcribed using random decamer primers and Superscript III Reverse Transcriptase (Invitrogen, Fisher Scientific, Denmark). Reactions without reverse transcriptase were performed in order to test for genomic amplification. Quantitative PCR was performed on an MX3000P system (Agilent, USA) based on Maxima Hot Start Taq DNA polymerase (ThermoFisher, Denmark). *Ptprg* mRNA levels relative to the reference genes *Rn18s* (18S ribosomal subunit) and *Actb* (β-actin) were evaluated in the different vascular beds based on the $2^{-\Delta\Delta C_T}$ method (*Livak and Schmittgen, 2001*). We used the following forward (F) and reverse (R) primers and probes (P) purchased from Eurofins Genomics (Germany): *Ptprg* (F: 5′ TGG TTA CAA CAA AGC GAA AGC CT 3′, R: 5′ ATA CTG ATC ACA CTT TCT CCT TCC 3′, P: 5′ ATC TGG GAA CAA AAC ACG GGA ATC ATC AT 3′), *Rn18s* (F: 5′ AAT AGC CTT CGC CAT CAC TGC 3′, R: 5′ GTG AGG TCG ATG TCT GCT TTC C 3′, P: 5′ TGG GGC GGA GAT ATG CTC ATG TGG TGT T 3′), and *Actb* (F: 5′ TGA CGT TGA CAT CCG TAA AG 3′, R: 5′ CTG GAA GGT GGA CAG TGA GG 3′ and P: 5′ AGT GCT GTC TGG TGG TAC CAC CAT GTA CC 3′). Probes were modified with 5′ 6-FAM and 3′ TAMRA. Each reaction consisted of 10 min at 95℃ followed by 50 cycles of 30 s at 95℃, 60 s at 55℃, and 60 s at 72℃.

### Single-cell RNA sequencing data

We explored levels of *Ptprg* transcripts in individual endothelial cells based on the EC Atlas database from VIB-KU Leuven, which is accessible at https://endotheliomics.shinyapps.io/ec_atlas/ (downloaded on 11 June 2020). This online database includes data from a recent single-cell RNA sequencing study on endothelial cells from healthy mice (*Kalucka et al., 2020*).

### Immunoblotting

We determined protein expression levels of eNOS in basilar and mesenteric arteries by immunoblotting using previously described antibodies (*Boedtkjer et al., 2011*; *Voss et al., 2019*). Arteries were snap frozen in liquid nitrogen and then homogenized using pellet pestles (Sigma-Aldrich, Denmark) in a lysis buffer at pH 7.5 containing (in mM) 20 Tris-HCl, 150 NaCl, 5 ethylene glycol tetraacetic acid (EGTA), 10 NaF, 20 β-glycerophosphate sodium salt, and HALT protease and phosphatase inhibitor

cocktail (Thermo Scientific, Denmark). Samples were sonicated for 45 s and centrifuged at ~16,000 g for 10 min. Total protein concentrations in the supernatants were measured using a bicinchoninic acid (BCA) protein assay kit (Thermo Scientific); and 10 µg total protein diluted in Laemmlie sample buffer (Biorad, Denmark) was loaded in each lane of an SDS polyacrylamide gel (Biorad). Membranes were first probed with anti-eNOS (RRID:AB_304967; 0.2 µg/mL ab5589; Abcam, UK) or anti-pan-actin (RRID:AB_2313904; 40 ng/mL #4968; Cell Signaling Technology, USA) primary antibody and then with secondary goat anti-rabbit antibody (RRID:AB_2099233; 30 ng/mL #7074; Cell Signaling Technology) conjugated to horseradish peroxidase. Bound antibody was detected by enhanced chemiluminescence (ECL Plus; GE Healthcare, Denmark) using an ImageQuant LAS 4000 luminescent image analyzer (GE Healthcare). Densitometric analyses were performed using ImageJ software (RRID:SCR_003070; Rasband; NIH, USA). Band densities of eNOS relative to pan-actin were reported after normalization to the average WT level.

## Blood pressure measurements

Mice were anesthetized by subcutaneous injection of ketamine and xylazine (80 mg/kg Ketaminol vet and 8 mg/kg Narcoxyl vet; Intervet International, The Netherlands) and placed on a thermostatically controlled heating platform.

For telemetry-based measurements of resting blood pressure and effects of L-NAME ingestion, the catheter of a telemetry transmitter (HD-X11; Data Sciences International, USA) was inserted in the common carotid artery through a midline incision in the neck, and the transmitter body placed in a subcutaneous pocket during stereomicroscopy. Pain relief was achieved through subcutaneous injection of buprenorphine (0.2 mL/kg, Temgesic, Schering-Plough, Europe). Telemetry measurements started one week after the operation. One 48 hr long registration at baseline was followed by two 72 hr long registrations during which first 0.5 mg/mL and then 3 mg/mL L-NAME was added to the drinking water. Telemetry signals were recorded for 10 s every minute using Dataquest A.R.T. 4.3 and analyzed with Ponemah 5.0 software (RRID:SCR_017107; Data Sciences International). We evaluated daytime and nighttime blood pressure from 11 AM to 1 PM and 11 PM to 1 AM, respectively.

For measurements of acute blood pressure responses to hyperventilation, a catheter was inserted in the common carotid artery through a midline incision and connected to a pressure transducer (MLT0699, ADInstruments). The mice were intubated and ventilated on a Minivent type 845 ventilator (Harvard Apparatus, USA) at a frequency of 125 min$^{-1}$ and with tidal volume adjusted until capnography readings (Capnograph Type 240, Hugo-Sachs Electronics, Germany) showed an $F_{ET}CO_2$ of 3.5%. During experiments, hypocapnia was induced by elevating the ventilation until $F_{ET}CO_2$ decreased to 2%. Mean arterial blood pressure was derived from the pressure traces using the blood pressure add-on for LabChart 8 Pro (ADInstruments). A few mice with an initial systolic blood pressure below 70 mmHg were excluded from the analysis.

## Laser speckle imaging

Mice were initially anesthetized by subcutaneous injection of ketamine (80 mg/kg) and xylazine (8 mg/kg) followed by injection with 1/4 the initial dose every 45 min. After endotracheal intubation, mice were ventilated under capnography control as described above. The head of the anesthetized mouse was fixed in an adaptor for a stereotaxic frame (World Precision Instruments, UK) while the rest of the body was kept warm on a heating pad (Fine Science Tools Inc, Canada). The skin covering the top of the skull was removed, the surface of the bone cleaned, and a coverslip mounted in agarose in order to minimize optical reflections. Whiskers were fixed to a metal pole mounted on a cylindrical solenoid controlled by an Arduino circuit (Funduino Kit, Germany). The whiskers on one side of the head were first stimulated by moving whiskers vertically with an amplitude of 8 mm and frequency of 4 Hz. Then, the stimulation protocol was repeated on the opposite side of the head, and the two responses were averaged for each mouse. Whisker stimulation was performed under control conditions ($F_{ET}CO_2$ = 3.5%) and after $F_{ET}CO_2$ was reduced to 2% by mechanical hyperventilation.

Speckle images of 1088 × 1088 pixels were captured with a Basler acA2000-165uc camera mounted on a VZM 200i Zoom Imaging Lens (Edmund Optics, USA) during transcranial illumination with near-infrared laser light (CLD 1011LP, Thorlabs Inc, USA). Speckle data were analyzed with MATLAB software (RRID:SCR_001622; MathWorks, USA). In each experiment, values were calculated

for a region of interest (100 × 100 pixels) in the second posterior bifurcation of the middle cerebral artery corresponding to the somatosensory barrel cortex contralateral to the whisker stimulation (*Aronoff and Petersen, 2008*).

## Arterial blood gas measurements

Arterial blood samples were collected through a catheter implanted in the common carotid artery of anesthetized, intubated mice ventilated to normocapnia ($F_{ET}CO_2$ = 3.5%) or experimental hypocapnia ($F_{ET}CO_2$ = 2%) on a Minivent type 845 ventilator (Harvard Apparatus). The blood was immediately analyzed using an ABL80 Flex blood gas analyzer (Radiometer, Denmark).

## Statistics of mouse experiments

Data are expressed as mean ± SEM and n equals number of mice (i.e. biological replicates). Probability (p) values less than 0.05 were considered statistically significant. Sample sizes were selected based on previous experience (*Boedtkjer et al., 2006*; *Boedtkjer et al., 2016c*; *Boedtkjer et al., 2011*; *Thomsen et al., 2014*) to allow detection of biologically relevant differences. If distributions were approximately Gaussian and variances equal between groups, we compared (a) one variable between two groups using unpaired two-tailed Student's *t*-test, (b) one variable between three or more groups using one-way ANOVA followed by Dunnett's post-tests, and (c) effects of two variables on a third variable using two-way ANOVA followed by Sidak's post-tests. If the distributions showed unequal variance (i.e. $p < 0.05$ by *F* test or by Brown-Forsythe and Bartlett's tests) or significant difference from normality (i.e. $p < 0.05$ by D'Agostino and Pearson or Shapiro-Wilk normality tests) due to right-skewness, we performed square root- or log-transformation. If variances were still unequal, we compared one variable between two groups based on unpaired two-tailed *t*-tests with Welch's correction. If data distributions still did not pass normality tests, we used the non-parametric Mann-Whitney test to compare one variable between two groups. Concentration-response relationships were fitted to sigmoidal functions using least-square regression analyses, and the derived log ($EC_{50}$), Hill Slope, and bottom values were compared using extra sum-of-squares *F*-tests. Investigators were not blinded for genotype during experiments. Data processing and statistical analyses were performed using Microsoft Office Excel 2016 (RRID:SCR_016137) and GraphPad Prism 7.05 (RRID:SCR_002798) software.

## Burden analysis of loss-of-function variants

We studied association between rare variation in *PTPRG* and human ischemic vascular disease based on exome sequencing data covering ~50,000 participants from the UK Biobank (RRID:SCR_012815), which is a cohort with deep genetic and phenotypic data collected from the United Kingdom (*Bycroft et al., 2018*; *Van Hout et al., 2019*).

Population-level variants (UK Biobank data field 23170) generated with the Functionally Equivalent pipeline (*Regier et al., 2018*) were used after ensuring that the genomic region containing *PTPRG* was unaffected by the current known error in the exome analysis protocol (UK Biobank Resource 3802). We predicted the consequences of alternative alleles within the full-length transcript for *PTPRG* (ENST00000474889.6) using the Ensembl Variant Effect Predictor (VEP) (*McLaren et al., 2016*); and included in our analyses, the 334 missense variants localized outside exon-intron boundary regions and 75 variants with predicted loss-of-function. Carriers of the identified variants were extracted using the R-package qgg (*Rohde et al., 2020*). The identified loss-of-function variants included 36 with low (intronic splice region variants, synonymous splice region variants), 17 with moderate (in-frame insertions, in-frame deletions, missense splice region variants), and 22 with high (splice acceptor variants, splice donor variants, stop-gain variants, frameshift variants) predicted impact. In total, we identified 29,975 individuals who carried at least one missense variant and 195 individuals, who carried a variant with predicted loss-of-function. Of the 195 loss-of-function carriers, 123, 32, and 40 individuals carried variants with low, moderate, and high predicted impact, respectively.

Known disease diagnoses for UK Biobank participants (data field 41270) were extracted from hospital inpatient records coded according to the International Classification of Disease, version 10 (ICD-10; RRID:SCR_010349). Amongst the 49,953 participants with exome data, 2529 individuals were diagnosed with angina pectoris (I20), 972 with acute myocardial infarction (I21), 392 with

cerebral ischemia (I63), and 153 with vascular intestinal disorders (K55.0). Self-reported health status was extracted for angina (data field 1074; reported by 1621 individuals), heart attack (data field 1075; reported by 1158 individuals), and stroke (data field 1081; reported by 686 individuals). We further extracted MRI-based values for 12,851 exome-sequenced individuals for whom left ventricular ejection fraction (data field 22420) was available from a fully automated analysis approach that has shown good correlation with manual analysis results performed by trained readers (*Suinesiaputra et al., 2018*). The cardiac MRI data set included 6482 carriers of missense *PTPRG* variants, 32 carriers of low impact loss-of function variants, 13 carriers of moderate impact loss-of-function variants, and 13 carriers of high impact loss-of-function variants.

We performed logistic regression analyses to calculate odds ratios (± SEM) for carrying missense or predicted loss-of-function *PTPRG* variants when diagnosed with ischemic vascular disease. We fist evaluated acute ischemic vascular disease as a whole for the brain, heart, and intestine (aggregate of ICD-10 diagnosis codes I20, I21, I63, and K55.0). For the group of individuals carrying a predicted loss-of-function variant of moderate or high impact, we then calculated separate odds ratios for I20, I21, and I63. The number of exome-sequenced participants in the UK Biobank diagnosed with K55.0 was too low for a meaningful separate statistical analysis. We next evaluated association of the *PTPRG* loss-of-function variants with self-reported angina, heart attack, and stroke. Finally, we evaluated the relation between carrier status for *PTPRG* variants and left ventricular ejection fraction by multiple linear regression analysis. The logistic as well as multiple linear regression analyses were corrected for sex (data field 31), age (data field 21022), body mass index (data field 21001), the first four genetic principal components (data field 22009), smoking status (data field 20116), dyslipidemia (ICD-10 diagnosis code E78.5), diabetes mellitus (ICD-10 diagnosis code E11), and hypertension (ICD-10 diagnosis code I10).

## Acknowledgements

The authors wish to thank Drs. Joseph Schlessinger (Yale University) and Walter F Boron (Case Western Reserve University) for generously providing the RPTPγ KO mice. We are grateful to Dr. Joanna Kalucka (Aarhus University) for assistance with the single-cell RNA sequencing analyses. We thank Viola Larsen, Liisa Jacobsen, Jørgen Andresen, and Jane Rønn (Aarhus University) for expert technical assistance. Access to the UK Biobank resource was approved under application number 60032.

## Additional information

### Funding

| Funder | Grant reference number | Author |
| --- | --- | --- |
| Det Frie Forskningsråd | 4183-00258B | Ebbe Boedtkjer |
| Det Frie Forskningsråd | 7025-00050A | Ebbe Boedtkjer |
| Lundbeckfonden | R93-A8859 | Ebbe Boedtkjer |
| Lundbeckfonden | R287-2018-735 | Palle D Rohde |
| MEMBRANES Research Center | | Ebbe Boedtkjer |

The funders had no role in study design, data collection and interpretation, or the decision to submit the work for publication.

### Author contributions

Kristoffer B Hansen, Palle D Rohde, Casper Homilius, Data curation, Formal analysis, Writing - review and editing; Christian Staehr, Sukhan Kim, Vladimir V Matchkov, Data curation, Writing - review and editing; Mette Nyegaard, Formal analysis, Writing - review and editing; Ebbe Boedtkjer, Conceptualization, Formal analysis, Supervision, Funding acquisition, Visualization, Methodology, Writing - original draft, Project administration, Writing - review and editing

Author ORCIDs
Palle D Rohde http://orcid.org/0000-0003-4347-8656
Vladimir V Matchkov http://orcid.org/0000-0002-3303-1095
Ebbe Boedtkjer https://orcid.org/0000-0002-5078-9279

Ethics
Human subjects: The research based on the UK Biobank resource was conducted under Application Number 60032.
Animal experimentation: This study was performed in accordance with the recommendations in the Guide for the Care and Use of Laboratory Animals of the National Institutes of Health. Protocols were approved by the Danish Animal Experiments Inspectorate (2016-15-0201-00982). All surgery was performed under general anesthesia, and every effort was made to minimize suffering.

## Decision letter and Author response
Decision letter https://doi.org/10.7554/eLife.57553.sa1
Author response https://doi.org/10.7554/eLife.57553.sa2

# Additional files

## Supplementary files
• Transparent reporting form

## Data availability
All data generated or analysed during this study are presented in the manuscript.

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
