## [Decision Letter]

**Acceptance summary:**

The manuscript describes the role of the receptor protein tyrosine phosphatase γ (RPTPγ) on vascular functions that are related to the sensing of acid-base disturbances. The main experiments are performed on RPTPγ knockout mice and complemented by an analysis of the correlation between *PTPRG* loss-of-function variations with ischaemic events at the population level.

**Decision letter after peer review:**

Thank you for submitting your article "*PTPRG* is an ischemia risk locus essential for HCO_3_^-^dependent regulation of endothelial function and tissue perfusion" for consideration by *eLife*. Your article has been reviewed by three peer reviewers, including Arduino A Mangoni as the Reviewing Editor and Reviewer #1, and the evaluation has been overseen by Matthias Barton as the Senior Editor.

The reviewers have discussed the reviews with one another and the Reviewing Editor has drafted this decision to help you prepare a revised submission.

Summary:

The manuscript describes the role of the receptor protein tyrosine phosphatase γ (RPTPγ) on vascular functions that are related to the sensing of acid-base disturbances. The main experiments are performed on RPTPγ knockout mice and complemented by an analysis of the correlation between *PTPRG* loss-of-function variations with ischemic events at the population level. The study is thoroughly performed and contributes significant knowledge to the field.

Essential revisions:

– Unusually, CO2/HCO_3_ buffer was prepared with Hepes. Such a dual buffer system provides high buffering power (twice the physiological). Can the authors clarify how the buffer was prepared? Was the pH of Hepes titrated before or after CO2/HCO_3_ addition, as this has a major effect on the equilibrium pH. It would seem important to include a 3-arm where the medium has CO2/HCO_3_ but no Hepes (i.e. to match buffering to Hepes-only).

– Hepes itself can produce effects on vessel contractility (PMID: 6254589); it would be important to confirm that the inclusion of Hepes did not affect RTPTγ signalling. Indeed, some early evidence suggested that Hepes blocks CAs (PMID: 1904692), and given the similarity to RTPTγ, such an interaction with endothelial targets should be tested. This could be tested in animals with CO_2_/HCO_3_ buffers that do not contain Hepes, and compare with Hepes (see also above).

– The removal of CO2/HCO_3_ buffer is likely to change intracellular pH (inactivation of pH regulators, drawing out an acidic gas). In the absence of a RTPTγ antagonist (for rescue), it seems important to check the degree to which cell pH is changing upon the buffer change, and to what degree this could explain the phenotype. This would have actions in both KO and WT, thus cannot alone explain the phenotype.

– The effect of SNP on NHE is likely not unique to this donor, and a similar effect would be attributed to other donors (via NO acting on NHE1). A more likely explanation seems that SNP can produce cyanide at high doses. As SNP may have these actions, it may be advisable to take this dataset out, as it is mechanistically inconclusive.

– When reporting results, the authors use present tense. I would recommend past tense because it emphasizes the historic nature of the observation.

– When hyperventilating mice, HCO_3_^-^ concentrations marginally dropped from 20.4 mmol/l to 19.0 mmol/l in wild-type mice. The authors claim that this small change could make the blood pressure RPTPγ-sensitive. This conclusion should be down toned.

– In hyperventilated mice, the blood pressure is lower in RPTPγ KO mice than in wild-type controls because the dilatory effect of RPTPγ is missing. If RPTPγ is a HCO_3_^-^ sensor, why does this effect become evident when HCO_3_^-^ concentrations are lowered? The same question arises for the experiment depicted in Figure 5F. The authors' conclusion seems to be based on some additional assumption that is not clearly stated.

– The finding on the neurovascular coupling is interesting. As this mechanism unlike hyperventilation comprises a local response and is limited to the brain area where neuronal activity occurs it would be interesting to know whether RPTPγ is also expressed in smaller arterioles or capillaries. The data of Kalucka et al., 2020, suggest that capillaries also express RPTPγ. If this is not true, the role of this enzyme in other cell types than endothelial cells like neurons or astrocytes during neurovascular coupling could be discussed.

– UK Biobank data analysis. The reported differences between cases and controls after age- and sex-matching is a useful initial step. However, the authors should conduct additional analyses after correcting for other clinical characteristics, particularly smoking, diabetes, hypertension, dyslipidemia, and history of previous cardiovascular events.

---

## [Author Response]

Essential revisions:– Unusually, CO2/HCO_3_ buffer was prepared with Hepes. Such a dual buffer system provides high buffering power (twice the physiological). Can the authors clarify how the buffer was prepared? Was the pH of Hepes titrated before or after CO2/HCO_3_ addition, as this has a major effect on the equilibrium pH. It would seem important to include a 3-arm where the medium has CO2/HCO_3_ but no Hepes (i.e. to match buffering to Hepes-only).

As we now explain in the manuscript (subsections “RPTPγ enhances endothelium-dependent vasorelaxation” and “Small vessel myography”), we included HEPES in the CO_2_/HCO_3_^–^-containing experimental solutions because we – being aware of previously reported unspecific effects – wanted to allow for comparison with experiments performed in CO_2_/HCO_3_^–^-free solutions, where the addition of an artificial buffer is unavoidable (see also the next point, below). In order to address the concern raised by the reviewers, we have now performed an additional series of experiments based on CO_2_/HCO_3_^–^-containing solutions without HEPES; and under these conditions, we corroborate that knockout of RPTPγ inhibits endothelium-dependent vasorelaxation (Figures 2C and 3C).

Regarding preparation of the experimental solutions, we adjusted pH after both HEPES and CO_2_/HCO_3_^–^ were added: after mixing all the ingredients, we heated the solutions to 37°C, bubbled aggressively with 5% CO_2_/balance air (HCO_3_^–^-containing solutions) or nominally CO_2_-free medical air (HCO_3_^–^-free solutions) – and then titrated the solutions to pH 7.40 while allowing sufficient time for the buffer systems to equilibrate. Although, this procedure is obvious to acid-base physiologists, we appreciate that *eLife* has a broader readership, and we have therefore now included these additional details in the manuscript (subsection “Small vessel myography”).

– Hepes itself can produce effects on vessel contractility (PMID: 6254589); it would be important to confirm that the inclusion of Hepes did not affect RTPTγ signalling. Indeed, some early evidence suggested that Hepes blocks CAs (PMID: 1904692), and given the similarity to RTPTγ, such an interaction with endothelial targets should be tested. This could be tested in animals with CO_2_/HCO_3_ buffers that do not contain Hepes, and compare with Hepes (see also above).

We acknowledge that HEPES has been reported to affect vessel contractility. In fact, as explained above, this was a main reason for including HEPES in both CO_2_/HCO_3_^–^-free and CO_2_/HCO_3_^–^-containing solutions thus allowing us to distinguish effects of omitting CO_2_/HCO_3_^–^ from potential effects of adding HEPES. In order to address the concern raised by the reviewers, we have now performed an additional series of experiments based on CO_2_/HCO_3_^–^-containing solutions without HEPES; and these data corroborate that also under conditions with no HEPES present, knockout of RPTPγ inhibits endothelium-dependent vasorelaxation (Figures 2C and 3C).

– The removal of CO2/HCO_3_ buffer is likely to change intracellular pH (inactivation of pH regulators, drawing out an acidic gas). In the absence of a RTPTγ antagonist (for rescue), it seems important to check the degree to which cell pH is changing upon the buffer change, and to what degree this could explain the phenotype. This would have actions in both KO and WT, thus cannot alone explain the phenotype.

This is indeed an important and relevant point, and we have now measured intracellular pH of endothelial cells in mesenteric and basilar arteries and evaluated the intracellular pH response to buffer change (Figure 5P and Q). We find that RPTPγ knockout does not influence endothelial steady-state intracellular pH, and we furthermore corroborate previous evidence from C57BL/6 mice (Boedtkjer et al., 2011; Boedtkjer, Damkier and Aalkjaer, 2012) demonstrating that omission of CO_2_/HCO_3_^–^ has no net effect on steady-state intracellular pH in endothelial cells. The effect of CO_2_/HCO_3_^–^ omission on steady-state pH_i_ depends on the relative activity of HCO_3_^-^ uptake (e.g., Na^+^,HCO_3_^–^-cotransporters) and HCO_3_^–^ extrusion (e.g., Cl^–^/HCO_3_^–^-exchangers) mechanisms – and varies amongst cell types and between mice on different genetic backgrounds. In mice on C57BL/6 genetic background, we consistently see no net steady-state difference in endothelial intracellular pH between experiments performed with and without CO_2_/HCO_3_^–^.

– The effect of SNP on NHE is likely not unique to this donor, and a similar effect would be attributed to other donors (via NO acting on NHE1). A more likely explanation seems that SNP can produce cyanide at high doses. As SNP may have these actions, it may be advisable to take this dataset out, as it is mechanistically inconclusive.

We appreciate this suggestion and have removed the data set with SNP (Figure 5).

– When reporting results, the authors use present tense. I would recommend past tense because it emphasizes the historic nature of the observation.

We have now changed the reporting of the results from present to past tense as suggested (see Results section).

– When hyperventilating mice, HCO_3_^-^ concentrations marginally dropped from 20.4 mmol/l to 19.0 mmol/l in wild-type mice. The authors claim that this small change could make the blood pressure RPTPγ-sensitive. This conclusion should be down toned.

Although our data support that systemic changes in [HCO_3_^–^] influence blood pressure, we appreciate that the observed changes in [HCO_3_^–^] are relatively small and have therefore followed the recommendation of the reviewers to tone down this conclusion (subsection “RPTPγ amplifies hyperventilation-induced blood pressure elevations”).

– In hyperventilated mice, the blood pressure is lower in RPTPγ KO mice than in wild-type controls because the dilatory effect of RPTPγ is missing. If RPTPγ is a HCO_3_^-^ sensor, why does this effect become evident when HCO_3_^-^ concentrations are lowered? The same question arises for the experiment depicted in Figure 5F. The authors' conclusion seems to be based on some additional assumption that is not clearly stated.

We have now included a discussion of how the unaltered resting blood pressure in RPTPγ KO mice (Figure 6A) most likely reflects that numerous mechanisms are involved in blood pressure control and that compensation—e.g., through nervous, hormonal or renal influences—can maintain blood pressure in the sustained phase and mask hemodynamic consequences of RPTPγ except when acute acid-base disturbances are imposed (Discussion, second paragraph).

– The finding on the neurovascular coupling is interesting. As this mechanism unlike hyperventilation comprises a local response and is limited to the brain area where neuronal activity occurs it would be interesting to know whether RPTPγ is also expressed in smaller arterioles or capillaries. The data of Kalucka et al., 2020, suggest that capillaries also express RPTPγ. If this is not true, the role of this enzyme in other cell types than endothelial cells like neurons or astrocytes during neurovascular coupling could be discussed.

Indeed the data of Kalucka et al., 2020, support that RPTPγ is broadly expressed in endothelial cells from arteries, capillaries, and the venous system. Based on the data from Kalucka et al. – that are publicly available through the EC Database at https://endotheliomics.shinyapps.io/ec_atlas/ – we have now expanded Figure 1 to show the endothelial expression patterns in different vascular beds and blood vessels of various sizes (Figure 1C-E).

– UK Biobank data analysis. The reported differences between cases and controls after age- and sex-matching is a useful initial step. However, the authors should conduct additional analyses after correcting for other clinical characteristics, particularly smoking, diabetes, hypertension, dyslipidemia, and history of previous cardiovascular events.

We thank the reviewers for these suggestions and have now thoroughly revised and updated the UK Biobank analysis. The following changes were done:

a) The initial genetic analysis of the exome variants within *PTPRG* was based on the SPB pipeline (see van Hout et al., 2019). It is now known that the SPB pipeline did not correctly identify genetic variants, which is why this data set has been retracted and replaced by a new genetic data set generated using the Functionally Equivalent pipeline (PMID: 30279509). For our study, this means that the number of genetic variants was reduced from 80 to 75 predicted loss-of-function (pLOF) variants in *PTPRG*.

b) Besides the errors within the original release of the UK Biobank exome data, we unfortunately observed another error relating to the extraction of individuals carrying pLOF variants. In our original report we stated that a total of 1,288 individuals were carrying at least one pLOF variant. This is not correct. We used an earlier version of the R package qgg (PMID: 31883004) to extract genotypes and phenotypes, which at that time had an issue reading the data correctly (such that individuals with missing genotypes were included). The package has been updated, and we have now extracted the correct information.

c) Since our first submission, seven UK Biobank participants have withdrawn their consent, and as a consequence we had to remove these individuals from our analysis.

d) In addition to these corrections, we have now also extracted additional information from the UK Biobank to allow us to conduct new analyses where we correct for sex, age, body mass index, genetic principal component, smoking status, dyslipidemia, diabetes mellitus, and hypertension. To best perform these analyses, we now test differences based on logistic regression analyses (Figure 7A-C) or multiple linear regression analyses (Figure 7D).

e) In the original analyses, we divided the UK Biobank participants between controls and carriers of any *PTPRG* pLOF variant irrespective of its predicted severity. In the new analysis, we have used additional information dividing the pLOF carriers into those with low, moderate, and high predicted impact.

f) We have now also expanded the analyses to include self-reported health status (Figure 7C) and left ventricular ejection fraction (Figure 7D).